# Molecular basis for the activation of outer dynein arms in cilia

Karim Housseini B. Issa[1], Muyang Ren[1], Bradley Burnet[2], Hao Lu[3,4], Charlotte Melia[1], Kate Heesom [2], Anthony J. Roberts [1], Sudipto Roy[3,4] & Girish R. Mali [1,2] ✉

Multiciliogenesis requires large-scale biosynthesis of motility-powering axonemal inner and outer dynein arm motors (IDAs and ODAs) before their intraflagellar transport (IFT) into cilia. ODAs are inhibited by the packaging chaperone Shulin during ciliogenesis in *Tetrahymena thermophila*. How Shulin is released for ODAs to become active inside cilia remains unclear. Here we uncover a molecular mechanism for ODA activation. We establish interactions between DNAAF9 (human Shulin) and mammalian ODA subunits, IFT proteins and the ciliary small guanosine triphosphatase (GTPase) ARL3 using proteomics and in vitro reconstitutions. Mutagenesis combined with biochemical and structural studies reveal that DNAAF9 and Shulin preferentially bind active Arl3–GTP highlighting a cross-species conservation of this interaction. GTP-loaded Arl3 can access, bind and displace Shulin from the packaged ODA–Shulin complex. We propose that, once the inhibited ODA complex enters growing cilia, Arl3–GTP displaces Shulin (DNAAF9) and sequesters it away from ODAs, promoting activation of their motility specifically inside cilia.

Motile cilia are microtubule-based extensions found on eukaryotic cell surfaces. Coordinated ciliary motion aids the locomotion of unicellular organisms, propels gametes and causes the essential flow of biological fluids over tissues in multicellular organisms. Ciliary beating is powered by axonemal outer dynein arm (ODA) motors and the waveform is modulated by inner dynein arms (IDAs)[1]. Ciliary dysmotility because of defects in ODAs causes primary ciliary dyskinesia (PCD), a severe human ciliopathy characterized by chronic respiratory symptoms[2].

ODAs and IDAs are macromolecular complexes that undergo cytoplasmic preassembly[3]. Large quantities of preassembled dyneins need to be transported to growing cilia and, because of their size, they are thought to mainly rely on the intraflagellar transport (IFT) system to enter cilia[4–6]. Following ciliary entry and transport to the growing tips, ODAs bind specific docking sites on doublet microtubules[7] (Fig. 1a). Throughout their trafficking from the cytoplasm to their final docking sites in cilia, ODAs need to be kept inactive to prevent off-target interactions. Tight control of ciliary targeting and the regulated activation

of dyneins is particularly important in mammalian multiciliated cells such as human airway epithelial cells (HAECs) or ciliated protozoa such as *Tetrahymena thermophila*, which deploy several thousands of preassembled ODA and IDA motors into several growing motile cilia. Premature cytoplasmic activation in such systems would have particularly deleterious consequences for the cells' survival.

Motor inhibition of ODAs is achieved by the packaging chaperone Shulin, which directly binds and locks them in a conformation that cannot productively engage with microtubules[8]. This inhibited ODA conformation is proposed to underpin their unimpeded ciliary import as it prevents aberrant interactions with cellular and ciliary microtubules during transport to ODA docking sites. The inhibited conformation of ODAs is similar to the compact inhibited conformation of dynein 2 on anterograde IFT trains[8,9]. Shulin and ODAs colocalize in regenerating *Tetrahymena* cilia, suggesting that it remains bound to ODAs to dampen their motor activity during ciliary transport. There are several outstanding questions: Is Shulin's inhibitory mechanism conserved in

[1]Sir William Dunn School of Pathology, University of Oxford, Oxford, UK. [2]School of Biochemistry, University of Bristol, Bristol, UK. [3]Institute of Molecular and Cell Biology (IMCB), Agency for Science, Technology and Research (A*STAR), Singapore, Singapore. [4]Department of Paediatrics, Yong Loo Ling School of Medicine, National University of Singapore, Singapore, Singapore. ✉e-mail: girish.mali@path.ox.ac.uk

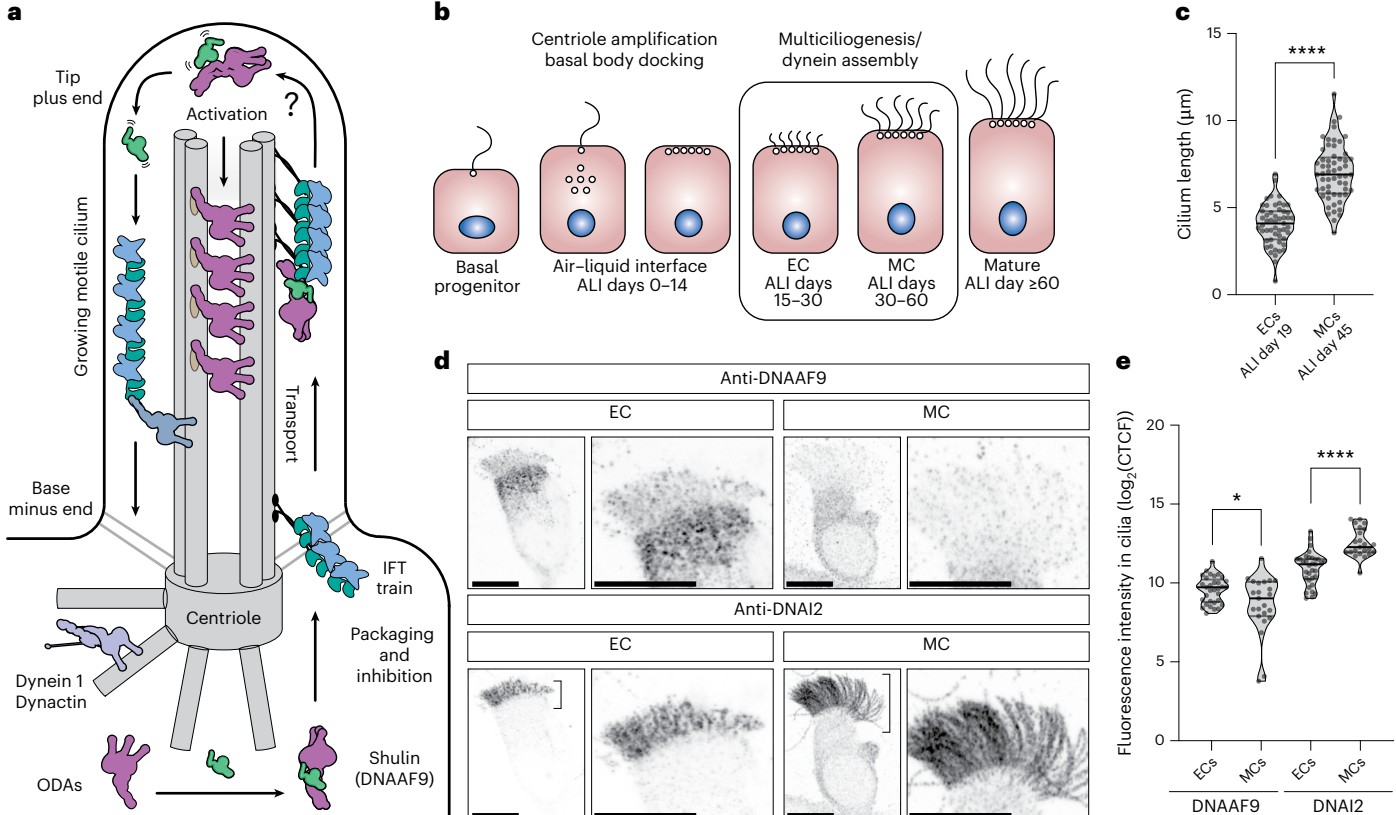

**Fig. 1 | DNAAF9 enters growing motile cilia in differentiating human airway cells. a**, Model of a motile cilium under construction involving Shulin/DNAAF9 for ODA packaging and inhibition. IFT trains transport ODAs to ciliary tips. How Shulin gets released for ODA activation inside cilia is an open question. **b**, Schematic of in vitro differentiation of HAECs undergoing multiciliogenesis at the ALI. **c**, Cilia length measurements from ALI day 19 (n = 53 ECs) and day 45 (n = 57 MCs) cells depicted as violin plots, with the means shown as black horizontal lines and s.d. as faint gray lines above and below the mean (mean ± s.d.: 7.0 ± 1.6 and 4.0 ± 1.1). An unpaired, two-tailed t-test was used to calculate P values. ****P ≤ 0.0001. **d**, Coimmunostaining of DNAAF9 and DNAI2

in an EC with short cilia (left) and an MC with long cilia (right). Representative images of cells from at least three replicate stainings are shown. Scale bars, 10 μm. **e**, Quantification of fluorescent signal intensities for DNAAF9 and DNAI2 in motile cilia of ECs (n = 33) and MCs (n = 23) depicted as violin plots, with the means shown as black horizontal lines and s.d. as faint gray lines above and below the mean (mean ± s.d.: DNAAF9 EC, 9.5 ± 0.8; DNAAF9 MC, 8.6 ± 1.8; DNAI2 EC, 10.9 ± 1.0; DNAI2 MC, 12.5 ± 0.9). Unpaired, two-tailed t-tests were used to calculate P values. *P ≤ 0.05 (P = 0.0145) and ****P ≤ 0.0001. Data in **e** were log-transformed ($log_2$(CTCF)).

its orthologs from other species such as its human ortholog DNAAF9? Does it interact with other transport factors to target packaged ODAs to cilia? Most importantly, how is Shulin released from ODAs to allow their final activation inside cilia?

In this work we determine a key function of Shulin (DNAAF9) in targeting ODAs to cilia beyond its inhibitory role and present a molecular mechanism for ODA activation. DNAAF9 in human airway cells coimmunoprecipitated with ODA complex subunits, IFT proteins and the ciliary small guanosine triphosphatase (GTPase) ARL3, which is essential for ciliogenesis and ciliary cargo trafficking. DNAAF9 binds and induces a closed conformation in ODAs purified from pig airway cilia. Furthermore, we show that Arl3 binds conserved surface exposed residues on the N1 domains of both DNAAF9 and Shulin in a GTP-dependent manner. Modeling Arl3 binding at Shulin's N1 domain and superimposing this model on the ODA–Shulin cryo-EM structure highlight steric clashes, which we suggest preclude the formation of a ternary complex involving ODA, Shulin and Arl3. In support of this notion, Arl3–GTP displaces Shulin from an inhibited ODA–Shulin complex in vitro. On the basis of our data, we propose a model where the inhibition imposed by Shulin on ODAs is relieved by Arl3 in its active GTP state. As there are higher pools of active Arl3 reported inside cilia[10,11], the displacement of Shulin and the resulting activation of ODA motor activity would, therefore, occur in a spatially restricted manner inside motile cilia.

## Results

### DNAAF9 enters growing airway multicilia and binds mammalian ODAs

Previously, Shulin's human ortholog DNAAF9 (C20ORF194) was pulled down with ARL3, which is a small GTPase that regulates the transport of both membrane and soluble ciliary cargo proteins[12–14]. This suggested to us that DNAAF9 could be involved in the ciliary cargo transport of ODAs. We performed AlphaFold2 (AF2) analyses, which revealed that DNAAF9 is structurally homologous to Shulin (Extended Data Fig. 1a–e), notably sharing domains critical for binding and inhibiting ODAs (N1 domain and C3 extension) (Extended Data Fig. 1f–h). On the basis of these observations, sequence conservation across evolution (Extended Data Fig. 2) and previous findings, we reasoned that DNAAF9 could package ODAs and cooperate with ARL3 to regulate their ciliary trafficking in diverse species.

To verify functional conservation, we investigated DNAAF9's functions using HAECs cultured at the air–liquid interface (ALI). HAECs grown in vitro differentiate from lung basal progenitors into multiciliated cells after exposure to air (airlift). After initial stages of centriole amplification and basal body docking (ALI days 0–14), cells progress asynchronously from a basal state through early–mid (ALI days 15–30) and then mid–mature (ALI days 30–60) stages to reach full maturity (>60 days) (Fig. 1b). Ciliary lengths define each differentiation stage[15].

ALI cultures at earlier stages typically contain a mixed population of basal, early-stage and mid-stage cells (here referred to as ECs) and start to grow short cilia (mean length: 4.03 ± 1.17 μm from $n$ = 53 cells). Later ALI cultures contain a mixture of early-stage, mid-stage and mature-stage airway cells (here referred to as MCs) and, on average, have comparatively longer cilia (mean length: 7.00 ± 1.6 μm from $n$ = 57 cells) (Fig. 1c). We reasoned that dynein assembly and trafficking would be most active at these stages of differentiation as cilia are still actively growing; therefore, we investigated the role of DNAAF9 using ECs and MCs.

First, we tracked DNAAF9's subcellular location in relation to the ODA intermediate chain (IC) DNAI2, which was used to mark cilia. Coimmunostaining ECs and MCs with anti-DNAAF9 and anti-DNAI2 antibodies revealed that the DNAAF9 signal concentrated apically near basal bodies and was also observed in short growing multicilia of ECs (Fig. 1d,e and Extended Data Fig. 3a). MCs showed more diffuse DNAAF9 signal in the cytoplasm and its levels were also slightly reduced from the longer MC cilia (Fig. 1d,e and Extended Data Fig. 3b). In agreement with the ciliary localization of DNAAF9 in human airway cells, two previous proteomics studies detected, at low abundance, the presence of DNAAF9 and its *Chlamydomonas* ortholog (Cre11.g467556.t1.1) in isolated mature respiratory airway cilia and wild-type full-length *Chlamydomonas* flagella[16,17] (Extended Data Fig. 3c,d). This suggests that DNAAF9 is able to enter the ciliary compartment at some stage of cilia biogenesis. The reduction in DNAAF9 levels in mature cilia compared to growing cilia with the strong enrichment at the base of airway cilia could mean that it undergoes rapid retrieval out of cilia for recycling in the apical cytoplasm. Importantly, DNAAF9's localization to growing human airway cilia resembles Shulin's expression in *Tetrahymena* cells undergoing ciliary regeneration[8]. Overall, our findings show that DNAAF9 is a ciliary localized protein in human respiratory cilia and *Chlamydomonas* flagella with a stronger enrichment in the growing motile cilia of human airway cells.

To gain molecular insights into DNAAF9's role during airway cell differentiation, we immunoprecipitated endogenous DNAAF9 from HAECs cultured to day 19 and day 45 after airlift. Coprecipitating proteins were identified by tandem mass tag (TMT) labeling followed by mass spectrometry (MS) proteomics. DNAAF9 was one of the top hits from cells at both differentiation time points. We detected subunits of the human ODA holocomplex (DNAH5, DNAH9, DNAI1 and LC3/NME9) amongst the proteins that coprecipitated with DNAAF9 (Fig. 2a,b). Western blot analysis of the immunoprecipitates (IPs) confirmed an interaction with the ODA IC DNAI2 (Fig. 2c).

To verify the association of DNAAF9 with mammalian ODAs more directly, we purified ODAs from pig tracheal cilia using sucrose density gradient fractionation and verified the presence of ODA subunits cosedimenting in the fractions by MS (Extended Data Fig. 4). Fraction 14 containing ODA heavy chains (HCs), ICs and light chains (LCs) (indicative of intact holocomplexes) was applied to an electron microscopy (EM) grid alone and after mixing with DNAAF9 and negatively stained. Intact double-headed ODA particles in isolation adopted the previously reported open 'active' conformation found in human respiratory cilia[18] (Extended Data Fig. 5a,c). Incubating DNAAF9 with fraction 14 causes a conformational change in pig ODAs from open to closed, similar to the closed, inhibited 'phi'-like state of triple-headed *Tetrahymena* ODAs when bound by Shulin[8] (Extended Data Fig. 5b,d,e).

Shulin binds the *Tetrahymena* ODA γ-HC Dyh3 at two sites[8]. Shulin's N1 domain makes contacts with multiple residues including the conserved LFGL motif in the γ-HC's tail helical bundles forming site 1. Shulin's C-terminal α-helical extension contacts conserved hydrophobic and acidic residues in γ-HCs motor near the AAA1(S) site. To model potential interactions between DNAAF9 and DNAH5 (human γ-HC), we performed AF3 (ref. 19) predictions using long sequences encompassing these sites. Structural predictions suggest potential contacts between DNAAF9 and DNAH5 encompassing the LFGL motif

and between the C3 extension and conserved residues near the AAA1(S) site (Extended Data Fig. 5f–i). Taken together, these findings indicate that human DNAAF9 interacts with mammalian ODAs and maintains them in a closed conformation, sharing its role as an inhibitor with its *Tetrahymena* ortholog Shulin.

## DNAAF9 interacts with proteins involved in cilia formation during airway cell maturation

In addition to ODA subunits, DNAAF9s EC and MC interactomes included the ciliary small GTPase ARL3, several IFT proteins and a few subunits of the cytoplasmic dynein 1 motor transport machinery (Fig. 2a,b). We performed Gene Ontology (GO) enrichment analysis to see whether any proteins amongst the top 87 common hits in both the EC and the MC interactomes clustered under specific GO terms. This highlighted an enrichment of IFT-B and ciliary tip proteins (Fig. 2d). We also identified proteins that uniquely coprecipitate with DNAAF9 in a cell-stage-specific manner but at lower abundance. The DNAAF9 interactome from ECs (mixture of basal, early and mid cell stages) contained more cytoplasmic proteins (DYNC1H1, DCTN2 and CAV1) in contrast to its interactome from MCs (mixture of early, mid and mature cell stages), which contained more centrosomal, basal body and axonemal proteins, including ciliary tip proteins (C1ORF87 and CCDC17), ODA docking complex subunits (ODAD1 and ODAD4) and microtubule inner proteins (CFAP57 and ENKUR)[20] (Fig. 2e). This observation suggests that DNAAF9's interactions change during airway cell multiciliogenesis, which could relate to its cellular functions in ciliary cargo trafficking.

To explore the functional role of DNAAF9, we obtained gross phenotyping data on *Dnaaf9*[−/−] mutant mice from the International Mouse Phenotyping Consortium (IMPC). These data revealed that 6.4% (6/94) homozygote animals developed severe postnatal hydrocephalus and had to be killed. Brain histopathology on four randomly selected homozygous animals that survived to 16 weeks revealed sublethal mild to moderate hydrocephalus compared to wild-type animals bred on the same C57BL6/J strain background (Extended Data Fig. 6a,b). Hydrocephalus is commonly associated with motile cilia defects in mice and frequently observed in other mouse models of PCD[21,22]. Additionally, in situ hybridization in zebrafish embryos showed that *dnaaf9* is broadly expressed with an enriched expression in the cranial regions at 24 hours after fertilization (Extended Data Fig. 6c). In vivo localization of C-terminally tagged Dnaaf9 protein showed its predominantly cytoplasmic localization in motile ciliated cells in the Kupffer's vesicle (the zebrafish left–right organizer) and the pronephric duct (kidney duct). Occasional weak signals were also detected in motile cilia of some cells (Extended Data Fig. 6d–f). We interpret this to mean that Dnaaf9 dynamically cycles between the cytoplasmic and ciliary compartments, in agreement with the notion that it is likely rapidly retrieved out of cilia to prevent aberrant inhibition of activated ODAs inside cilia. These phenotypic observations and in vivo localization studies combined with the proteomic links to ciliary proteins support DNAAF9's role in the formation of functional motile cilia.

Using quantitative TMT proteomics, we next compared the relative abundances of top DNAAF9 protein interactors between ECs and MCs. This revealed that DNAAF9 levels reduce as airway cells mature, which matches the overall reduction in DNAAF9 signals observed by immunostaining of MCs. This could be because of a downregulation in its gene expression or its enhanced protein degradation in the cell (Fig. 2f). DNAAF9 coprecipitates relatively higher levels of cytoplasmic dynein 1 LC (DYNC1LI2) and IC (DYNC1I2) subunits from ECs compared to MCs. Conversely, relatively higher levels of ARL3 coprecipitate with DNAAF9 from MCs compared to ECs (Fig. 2f). As several IFT-B subunits coprecipitated with DNAAF9 from both ECs and MCs (Fig. 2a,b) and IFT-B is a significantly enriched GO term, we verified two of the top IFT-B protein hits, IFT74 and IFT81, as putative interactors of DNAAF9 by IP western blot analyses (Fig. 2g,h).

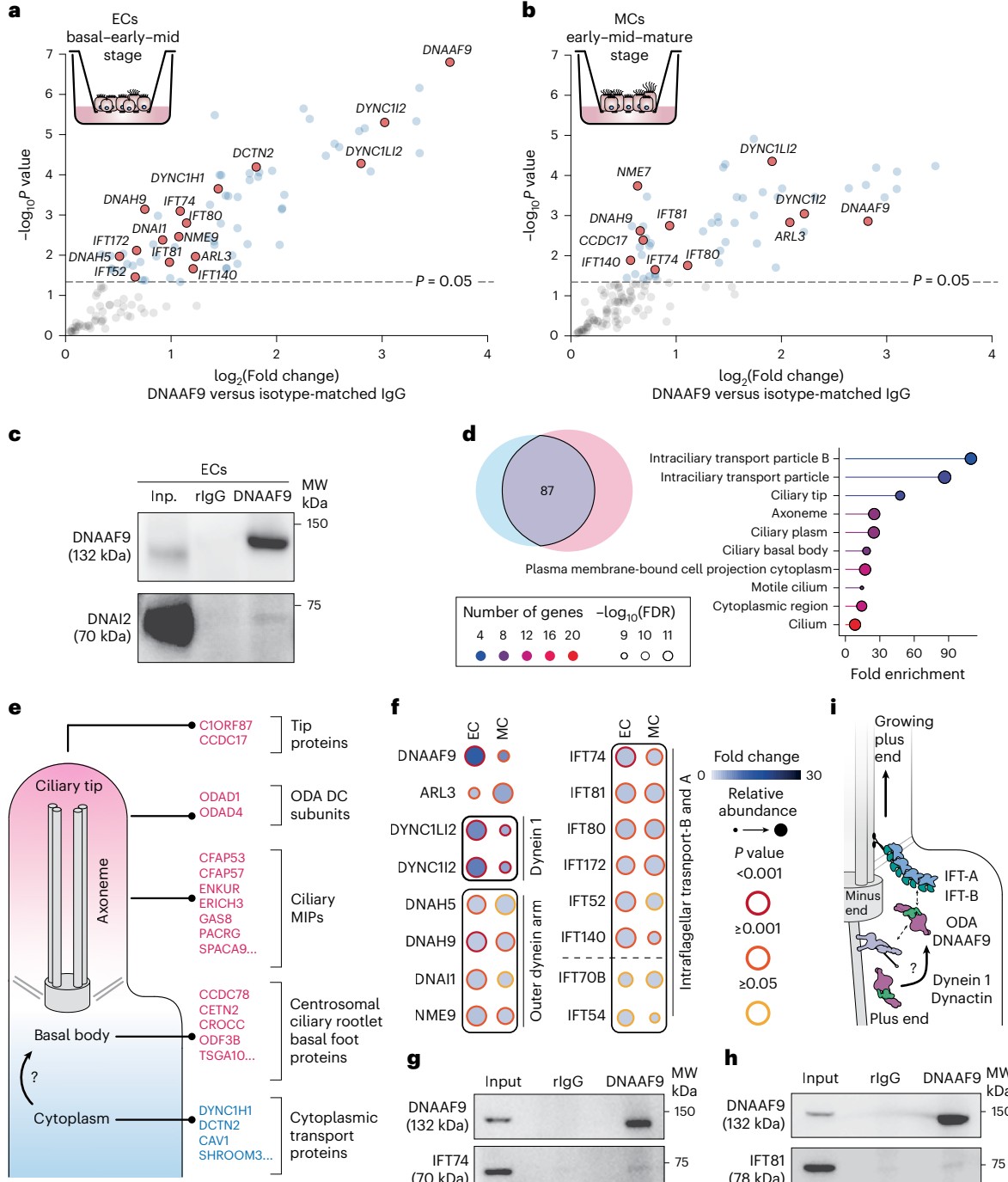

**Fig. 2 | DNAAF9 interacts with ODA subunits and multiple transport proteins in human airway cells. a,b**, DNAAF9 interactome from ALI ECs on day 19 (**a**) and MCs on day 45 (**b**). Proteins are plotted by enrichment (DNAAF9 IP versus isotype-matched IgG IP) on the x axis and significance (n = 3 experimental replicates) on the y axis. P values were derived using a two-tailed t-test with unequal variance for the DNAAF9 IP versus isotype-matched IgG IP samples. Proteins with P values < 0.05 (−log₁₀1.3) are shaded in gray, key hits are labeled and highlighted with pink circles, and all other proteins in the datasets are shaded in blue. **c**, A coimmunoprecipitation western blot image for DNAAF9 and DNAI2 from ECs representative of two repeats is shown. MW, molecular weight markers in kDa. **d**, Venn diagram of 87 overlapping proteins from DNAAF9's EC (blue) and MC (pink) interactomes. The dot plot shows the GO term enrichment of cellular component for the overlapping hits determined using ShinyGO with enriched terms (y axis) ranked by fold enrichment (x axis) over a background set of genes (FDR cutoff = 0.05). The points are colored by number of proteins clustering within each term and sized by enrichment confidence (FDR). **e**, Proteins that exclusively coprecipitate with DNAAF9 from ECs (in blue) or MCs (in pink) are grouped according to their subcellular location. **f**, Fold change and relative abundances of key DNAAF9 interactors from ECs and MCs are compared in a dot plot generated using ProHits-viz. The points are colored in shades of blue according to the protein fold change and sized by relative abundance. The color of the outline represents significance. **g,h**, Coimmunoprecipitation western blot images for DNAAF9, IFT74 (**g**) and IFT81 (**h**) from ECs representative of two repeats. **i**, Schematic of a growing cilium summarizing the key putative interactors of DNAAF9 along the ODA trafficking route.

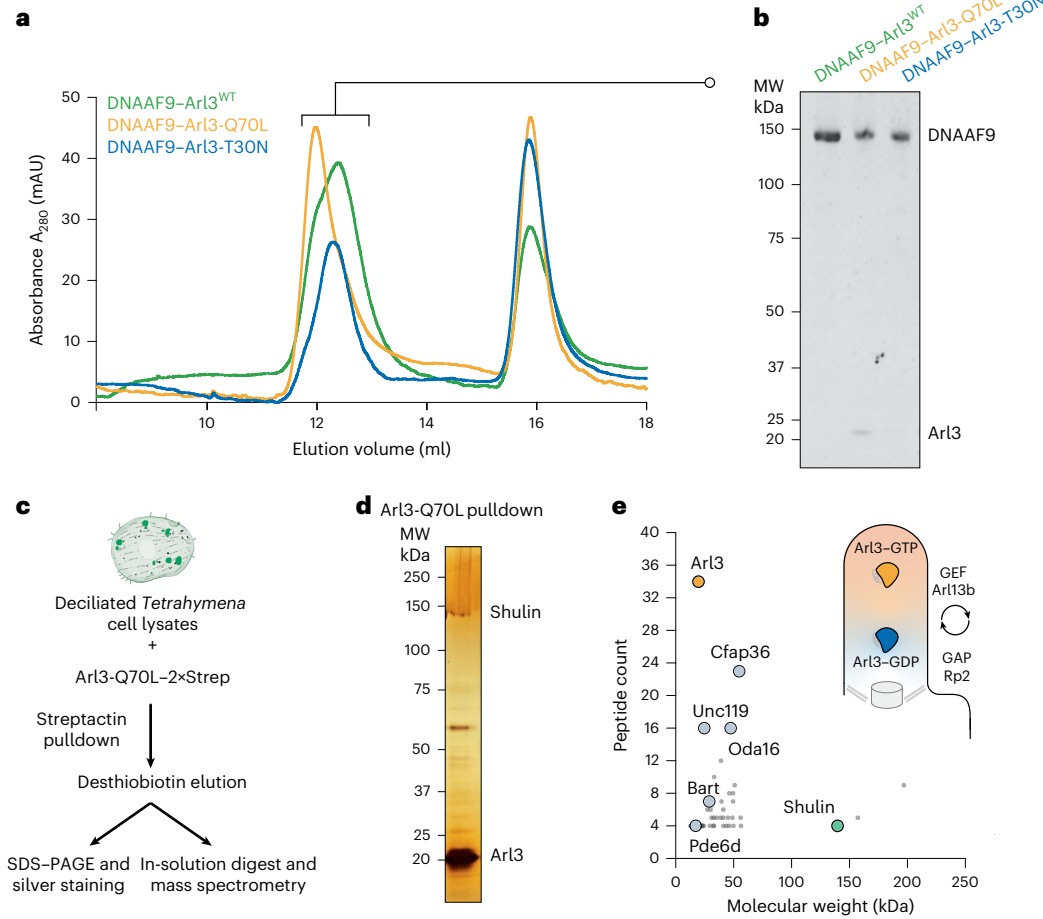

**Fig. 3 | Arl3-Q70L coelutes with DNAAF9 and pulls down endogenous Shulin from growing motile cilia. a**, Analytical size-exclusion chromatography of DNAAF9–Arl3 complexes (three Arl3 variants: wild type and the GTP (Q70L) and GDP (T30N) mimicking mutants). **b**, Gel image of proteins eluting in the first peak from each of the three reconstitutions. **c**, Scheme used to identify interactors of activated Arl3 from deciliated *Tetrahymena* cells undergoing ciliary regeneration using the GTP-locked Q70L variant. **d**, Silver-stained gel from a single pulldown showing proteins coeluting with Arl3-Q70L. **e**, Scatter plot showing a graphical representation of the top hits pulled down by Arl3-Q70L identified by MS ranked by peptide count (*y* axis) versus molecular weight (*x* axis). Key hits are highlighted: Arl3 (mustard circle), Shulin (green circle) and other known effectors (gray circles). All other proteins in the dataset are shaded as small gray dots. The graph corresponds to the silver-stained gel in **d**. The inset schematic shows Arl3's GTPase cycle. Active Arl3–GTP is enriched inside cilia by its ciliary GEF Arl13b and inactive Arl3–GDP is excluded as its GAP Rp2 localizes to the basal body.

Although, the precise site where ODAs bind IFT trains remains poorly resolved, on the basis of our proteomics evidence here and analyses of recent IFT structures[23,24], we reasoned that ODAs could bind through the peripherally located IFT81 and IFT74 subunits within the IFT-B1 sub-module in anterograde trains. Reconstituting the purified *Tetrahymena* ODA–Shulin complex with a purified human IFT74–IFT81 complex followed by immunoblotting revealed that all components coeluted in the complex peak indicating binding (Extended Data Fig. 7a–c). These data support the notion that the coupling of ODAs packaged by Shulin to the IFT system can occur through the IFT74 and IFT81 subunits.

On the basis of the DNAAF9 protein interaction data, we propose the following sequence of events (Fig. 2i). First, DNAAF9 binds ODAs in the cytoplasm. Presently, it is unclear whether it binds ODAs cotranslationally or after they are fully preassembled. ODAs then reach the peribasal body pool. Apical enrichment around the basal body could be achieved through the dynein 1 transport machinery or through another mechanism. More work is needed to address this. The apically enriched pool of ODAs is maintained in an inhibited state by DNAAF9 to facilitate crossing the diffusion barrier at the transition zone through active transport on IFT trains by coupling to the IFT74 and IFT81 subunits.

## Arl3–GTP binds DNAAF9 in vitro and pulls down Shulin from growing *Tetrahymena* cilia

Once inside cilia, DNAAF9/Shulin must detach from ODAs to allow their activation and stable axonemal incorporation for ciliary beating. Shulin's release mechanism remains unknown. According to coarse-grained molecular dynamics simulation of the ODA–Shulin complex, it has been proposed that conformational remodeling during ODA docking to doublet microtubules is sufficient to release Shulin[25]. However, these simulations were performed after decreasing the attractive forces between Shulin and ODA subunits to observe the detachment. We, therefore, reasoned that additional factors specifically inside cilia could first destabilize the ODA–Shulin complex, triggering its release upon microtubule association. As ARL3 is an established ciliary cargo release factor that interacts with DNAAF9 in airway cells, we hypothesized that its binding to DNAAF9 could trigger its detachment from ODAs inside cilia. First, we probed this interaction by reconstituting DNAAF9 with the *Tetrahymena* ARL3 ortholog, Arl3, in its wild-type form and its GTP and guanosine diphosphate (GDP) mimicking states using the Q70L and T30N variants, respectively, for in vitro reconstitutions (Fig. 3a,b). In vertebrates, ARL3-Q71L is described as a dominant active variant because of defective GTP hydrolysis and conformationally mimics the constitutive GTP-loaded active state. ARL3-T31N is

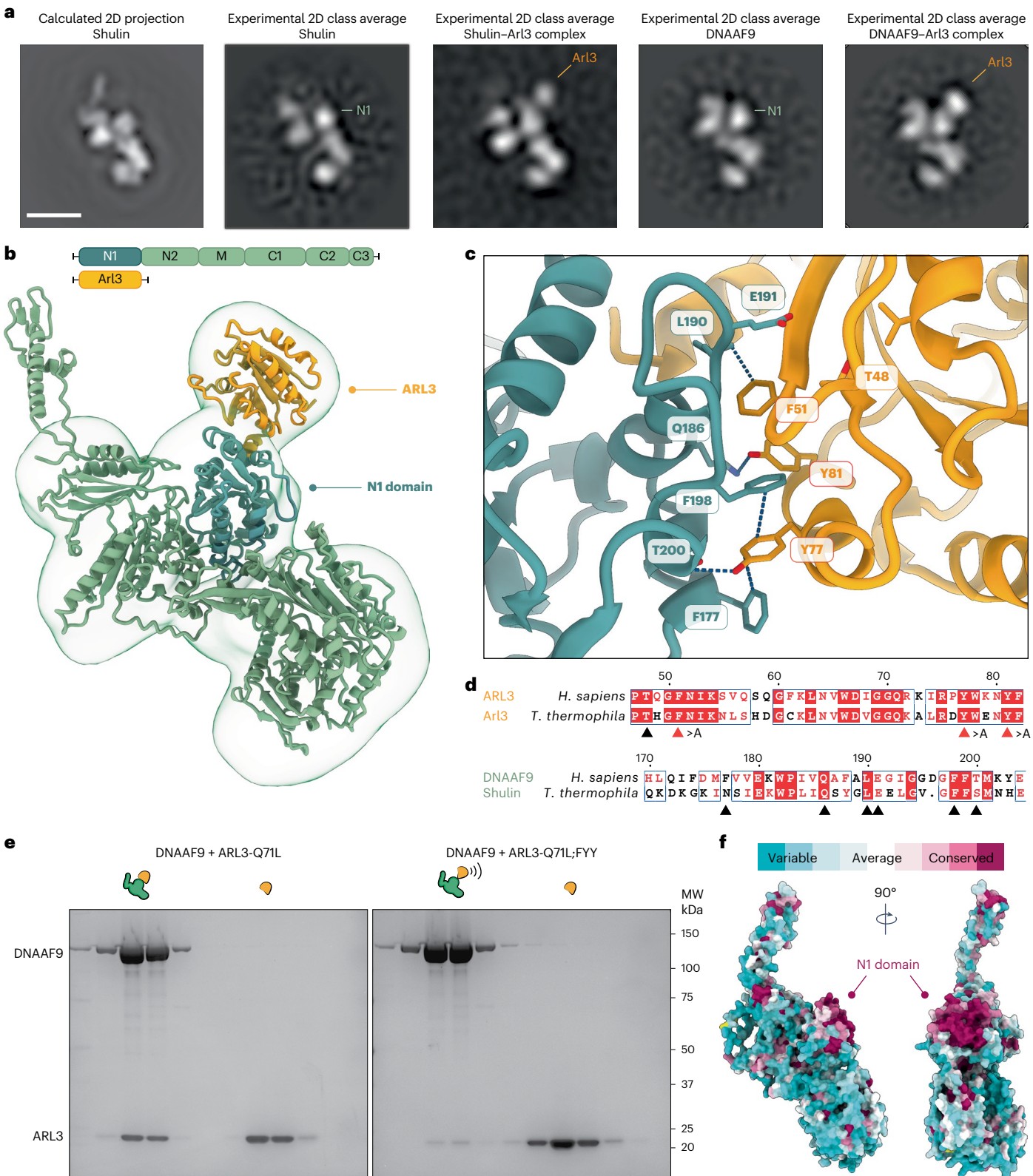

**Fig. 4 | ARL3/Arl3 binds DNAAF9/Shulin at its conserved N1 domain through aromatic and polar residues. a**, Side view of a calculated 2D projection of Shulin (PDB 6ZYX_8) and best-matching 2D class averages for DNAAF9/Shulin alone and bound by Arl3-Q70L. The N1 domain and position of Arl3 are highlighted. Scale bar, 2.5 nm. **b**, AF2-Multimer prediction of the DNAAF9–ARL3 complex rigid-body docked into a negative-stain EM density map of the complex. The cartoon highlights the domain organization **c**, AF2-Multimer-predicted binding interface between the DNAAF9 N1 domain and ARL3, with residues forming interactions (dashed bonds) labeled. **d**, Sequence alignment of human and *Tetrahymena* DNAAF9/Shulin and ARL3/Arl3, with interface residues marked by arrowheads. Positions of single residues substituted with alanine (>A) residues for complex disruption are marked by red arrowheads. **e**, Analytical size-exclusion chromatography fractions from reconstitutions to test complex formation between DNAAF9 and ARL3-Q71L or ARL3-Q71L;FYY. **f**, CONSURF scores mapped onto DNAAF9's predicted structure. The N1 domain comprising the conserved surface patch where ARL3 binds is highlighted.

the dominant negative variant that is defective in GTP binding and is thought to conformationally mimic the GDP-loaded inactive state[13,26]. A pool of active ARL3 is maintained inside cilia because of the spatially restricted localizations of its guanine nucleotide exchange factor (GEF) and GTPase-activating protein (GAP): ARL13B on the ciliary membrane and RP2 at the ciliary base, respectively[27–29]. Reconstitutions showed that only Arl3-Q70L coeluted with DNAAF9 even in the presence of GDP (Fig. 3a,b), likely because of its stable GTP-locked conformation. These cross-species reconstitutions also indicated that the DNAAF9–ARL3 complex is evolutionarily conserved.

DNAAF9 (C20ORF194) was previously reported to be pulled down by active ARL3-Q71L from primary ciliated human RPE-1 cells[13]. Our proteomic data showed that both proteins coprecipitated in differentiating human airway motile multiciliated cells. To test this interaction more directly in the context of actively ciliating cells, we performed a pulldown using recombinant *Tetrahymena* Arl3-Q70L as bait and lysates of *Tetrahymena* cells undergoing ciliary regeneration after deciliation (Fig. 3c). Analysis of the coprecipitating prey proteins by MS identified several known effectors of Arl3 (Cfap36/Bartl1, Unc119 and Pde6d)[28,30] and Shulin (Fig. 3d,e).

We conclude that both human DNAAF9 and *Tetrahymena* Shulin preferentially bind the GTP-locked active variants of ARL3 and Arl3, respectively. Our findings, in the context of previous data, highlight the cross-species conservation of this interaction that occurs in both primary and motile ciliated cells. Overall, we establish DNAAF9 (Shulin) as an effector of ARL3 (Arl3) that functions under specific contexts, such as during motile multiciliogenesis, likely to regulate the transport of ODAs to cilia.

## ARL3 binds DNAAF9's N-terminal domain

Next, we sought to visualize where ARL3 bound to DNAAF9 to ascertain whether its binding could explain DNAAF9's and Shulin's detachment from ODAs. We performed negative-stain EM on purified recombinant DNAAF9 and Shulin in complex with Arl3-Q70L. Single-particle analysis provided well-defined two-dimensional (2D) class averages (Fig. 4a and Extended Data Fig. 8a,b). Comparing the averages to a calculated 2D projection profile of Shulin (Protein Data Bank (PDB) 6ZYX_8) revealed the presence of an extra density adjacent to the N-terminal N1 domain of DNAAF9/Shulin. This density was absent in about half of the class averages. We conclude that these class averages represent DNAAF9 and Shulin alone and in complex with Arl3 that binds the N1 domain.

We next performed AF2-Multimer predictions to generate a structural model of the DNAAF9–ARL3 complex using full-length wild-type human sequences. The model predicted binding of ARL3 in a GTP-loaded conformation to the N1 domain of DNAAF9, in agreement with the 2D class averages, and could be accommodated into an 18-Å three-dimensional (3D) density map of the DNAAF9–Arl3 complex obtained by negative-stain EM (Fig. 4b and Extended Data Fig. 8c–e). Closer inspection of the binding region revealed a string of hydrogen bonds and π–π stacking interactions spanning the interface between conserved residues from DNAAF9s N1 domain and ARL3's conserved N-terminal half (Fig. 4c,d and Extended Data Fig. 8f). We reasoned that substituting interface residues on an ARL3-Q71L background could destabilize the complex and verify the predicted model. We generated an ARL3 triple mutant (FYY) substituting three conserved interface aromatic residues (F51, Y77 and Y81) with alanine's on the GTP-locked Q71L background (Fig. 4d). Reconstitutions under saturating GTP (1 mM) conditions showed that the ARL3-Q71L;FYY triple mutant had a highly reduced affinity for DNAAF9 compared to the ARL3-Q71L GTP-locked variant (Fig. 4e). Lastly, we performed CON-SURF analysis and mapped per-residue conservation scores on the predicted DNAAF9 structure. This revealed a highly conserved patch of surface exposed residues in the N1 domain in addition to a few conserved residues at the tip of the C3 extension that contact the motor domain (Fig. 4f).

To understand the nucleotide dependent binding of ARL3 to DNAAF9, we aligned our predicted model with previous X-ray structures of murine ARL3 in the GTP-locked and GDP-locked states (PDB 4ZI2 and PDB 1FZQ, respectively). Several structural features in ARL3 (α3 helix, interswitch β2–β3 strands and L3 loop) participate in the interaction with DNAAF9 (Extended Data Fig. 9a). In the GDP-loaded state, the L3 loop and β2 strand reconfigure to form a β-hairpin, which displaces the interacting α3 helix and sterically clashes with DNAAF9's N1 domain (Extended Data Fig. 9b). We also performed AF3-Multimer predictions between DNAAF9 and ARL3 to model the complex in the presence of GTP and GDP. AF3 generated a high-confidence model for the interaction between GTP-loaded ARL3 and DNAAF9, positioning it at the N1 interface, but failed to model an interaction with GDP-loaded ARL3 (as evidenced by a lack of residue-level pairwise association in the predicted aligned error (PAE) plots), likely because of steric clashes (Extended Data Fig. 9c,d).

Validating the structural modeling, reconstitution experiments between human DNAAF9 and ARL3, as well as *Tetrahymena* Shulin and Arl3, using the QL and TN variants under saturating GDP and GTP conditions also showed that DNAAF9 and Shulin bound the QL (GTP-mimic) variant more strongly than the TN (GDP-mimic) variant (Fig. 3a,b and Extended Data Fig. 10). Similar to reconstitutions between human DNAAF9 and ARL3-Q71L;FYY or ARL3-Q71L (Fig. 4d and Extended Data Fig. 10a,c,d,f,m), reconstitutions between Shulin and Arl3-Q70L;FYY or Arl3-Q70L showed that the interface triple mutant has highly reduced binding to Shulin compared to the Arl3-Q70L variant (Extended Data Fig. 10g,i,j,l,n). We conclude that substituting the three aromatic residues destabilizes the interactions at the interface for the complex to be stably formed. Overall, our analyses support the idea that the N1 domain harbors the binding site for GTP-locked ARL3 (Arl3) on DNAAF9 (Shulin).

## Active Arl3 binds and displaces Shulin from ODAs

We hypothesized that the binding of active Arl3 to Shulin would displace it from packaged (inhibited) ODAs and sequester it away from open (activated) ODAs to prevent rebinding. To test this directly, we first reconstituted *Tetrahymena* ODAs purified from cilia with recombinant Shulin, which induces the 'phi'-like closed conformation[8] (Fig. 5a). We then performed in vitro displacement assays by equally dividing the purified ODA–Shulin complex and incubating it, under saturating GTP (1 mM) levels, with an excess of either the GTP-locked Arl3-Q70L variant or the Arl3-Q70L;FYY mutant that is unable to strongly bind Shulin. We resolved the resulting complexes by size-exclusion chromatography (Fig. 5b,c). Quantification of band intensities (gel densitometry) in the peak fractions showed that the level of Shulin coeluting with ODA holocomplex subunits (HCs and IC2) is higher in the presence of Arl3-Q70L;FYY compared to Arl3-Q70L (Fig. 5d–f). We conclude that adding active Arl3 to the purified ODA–Shulin complex has a destabilizing effect and displaces most of the bound Shulin. We suggest that the formation of a Shulin–Arl3 complex sequesters free Shulin, preventing its rebinding to ODAs. The Arl3 mutant that is unable to bind Shulin does not have a similar destabilizing effect on the ODA–Shulin complex.

We next performed structural modeling to see whether it matches our biochemical observations. Arl3 and ODAs contact Shulin's N1 domain at distinct binding sites. Although there is no direct overlap between these sites, superimposing the Shulin–Arl3 prediction onto the cryo-EM-guided structure of ODA–Shulin highlighted steric clashes between Arl3 and ODA HCs (Fig. 6a,b). Taken together, the displacement assays and structural studies provide a molecular explanation for how clashes upon binding of active Arl3 would destabilize the ODA–Shulin complex, ultimately promoting the release and sequestration of Shulin. Overall, we propose a model where, by displacing Shulin, Arl3 relieves the inhibition imposed on ODAs, allowing the motors to undergo structural reconfigurations required for the activation of motility (Fig. 6c).

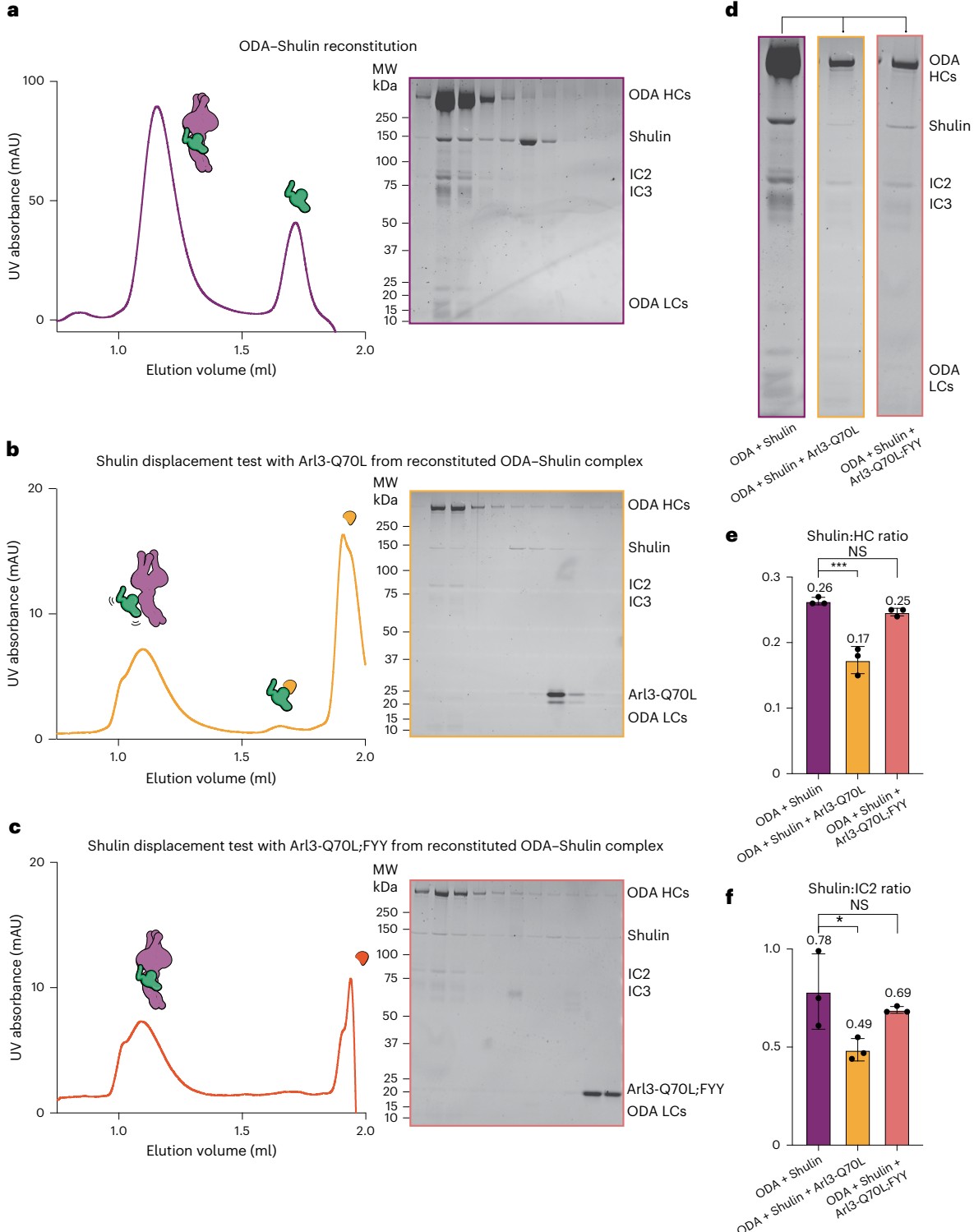

**Fig. 5 | Active Arl3 displaces Shulin from ODAs. a–c**, Analytical size-exclusion chromatography traces and corresponding gels showing biochemical reconstitution and displacement assays to test the displacement of Shulin from a purified ODA–Shulin complex (**a**) by Arl3-Q70L (**b**) or Arl3-Q70L;FYY (**c**). **d**, Peak fraction from reconstituted ODA–Shulin complex in **a**. This fraction was divided into equal halves and incubated in saturating GTP conditions (1 mM) with an excess of the GTP-locked Arl3-Q70L variant (**b**) or the Arl3-Q70L; FYY triple mutant (**c**). Peak fractions from these displacement assays are shown.

**e**,**f**, Shulin:HC (ODA HCs) and Shulin:IC2 ratios calculated from gel densitometry analyses (band intensities) on triplicate runs for each experimental condition (ODA–Shulin reconstitution and displacement assays in the presence of Arl3-Q70L or Arl3-Q70L;FYY) are plotted. The mean ratio values are shown above each column. A one-way ANOVA with Tukey's test for multiple comparisons was used to derive $P$ values. NS (not significantly different; $P = 0.323$) and ***$P \leq 0.001$ ($P = 0.0003$) (**e**); NS ($P = 0.613$) and *$P \leq 0.05$ ($P = 0.0464$) (**f**). Error bars represent the mean and s.d.

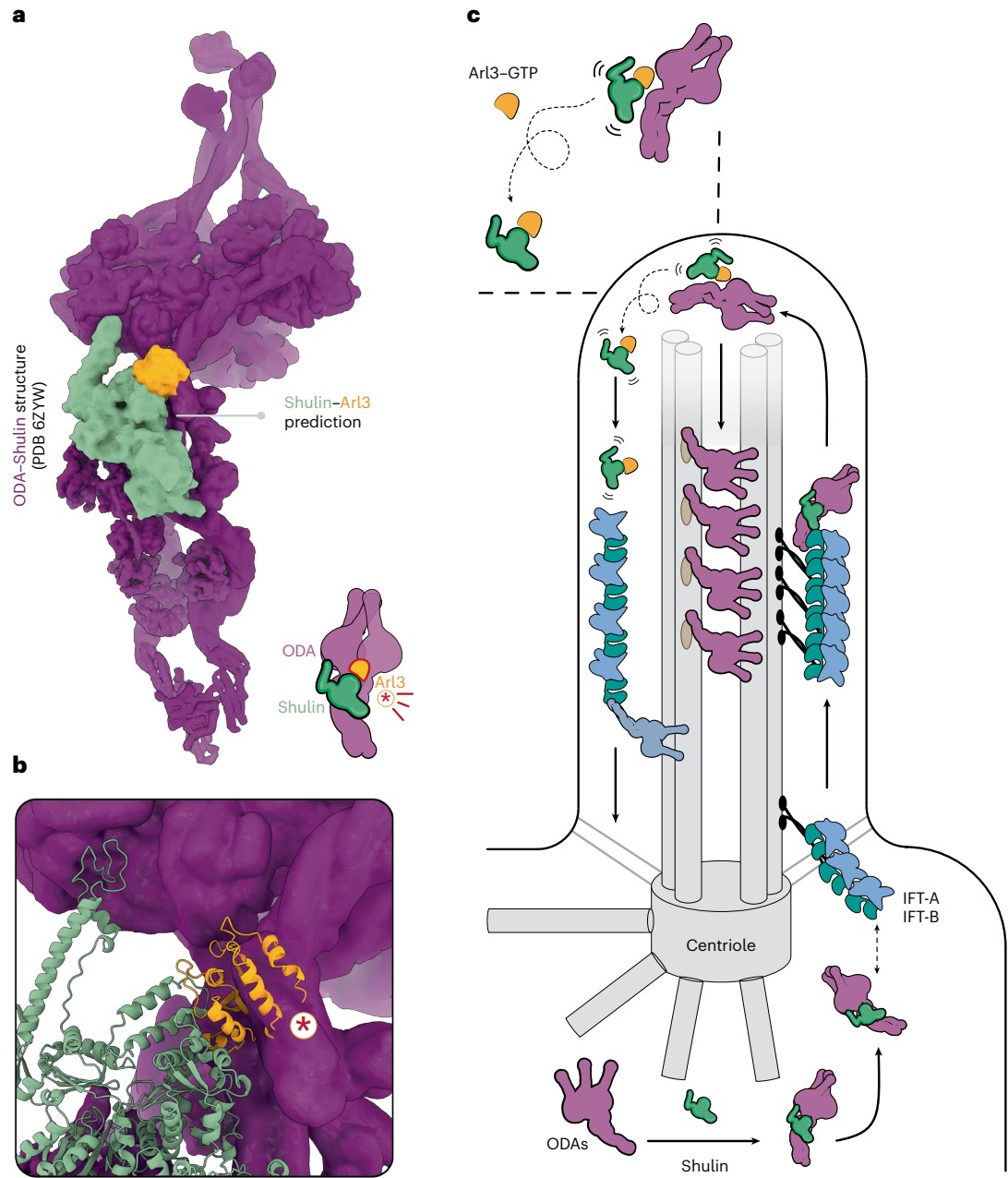

**Fig. 6 | Model of Shulin release by Arl3 for ODA activation inside cilia.**
**a**, Shulin–Arl3 prediction generated using AF superimposed on the experimental structure of ODA–Shulin complex (PDB 6ZYW). **b**, Structural model and cartoon representation highlighting the steric clashes (encircled asterisk) between Arl3 and ODAs that would prevent the formation of a stable ODA–Shulin complex. **c**, Proposed model highlighting Shulin's role in delivering ODAs to motile cilia.

Shulin locks ODAs in a closed conformation and chaperones their attachment to IFT trains near the basal body for ciliary entry. Inside cilia, Shulin is displaced by active Arl3–GTP for ODA activation and stable axonemal incorporation. Shulin could be sequestered in a complex with Arl3 that exits cilia for recycling or degradation in the cytoplasm.

## Discussion

Our work provides a molecular explanation of how ODAs get activated in cilia. We show that the packaging chaperone Shulin is displaced from *Tetrahymena* ODAs by the ciliary GTPase Arl3 to promote motor activation. On the basis of the conservation of Arl3's interaction with Shulin and its human ortholog DNAAF9, we propose that the mechanism of ODA activation would be conserved in several ciliates.

DNAAF9's interaction with ARL3, a ubiquitously expressed GTPase, suggests that, in metazoans, it could be important for functions

beyond the ones we define here. The DNAAF9–ARL3 interaction was first detected in retinal RPE-1 cells that have primary cilia[13]. Although DNAAF9 variants have not been reported in persons with ciliopathies, its broad expression pattern in humans and interaction with ARL3 across different tissues indicate that it could have critical roles in tissues with both primary and motile cilia. Indeed, *Dnaaf9*[−/−] mouse mutants develop mild to severe hydrocephalus, consistent with a role in contributing to proper motile cilia functions. Taken together, we propose that DNAAF9 may be functionally important in maintaining

ciliary motility in mammals and should be considered as a candidate for syndromic ciliopathies with combined features characteristic of primary and motile cilia defects.

We determine another key molecular function for DNAAF9 (that is, ODA transport in human airway cells). On the basis of its interactions and subcellular concentration at the base of growing multicilia, we speculate that DNAAF9 could chaperone the attachment of packaged ODAs to IFT trains in multiciliated cells. We propose that the coupling of packaged ODAs to IFT could be mediated by the IFT74–IFT81 dimer. Further work is needed to decipher the precise molecular mechanism of ODA–IFT coupling.

We extend ARL3's functions to include the regulation of ODA transport and their timely activation in motile cilia. Complete loss of ARL3 is lethal in vertebrates[31] and loss-of-function ARL3 variants have been classically linked to severe primary ciliopathies because of perturbed trafficking of ciliary signaling receptors[12,32]. Arl3 is essential for the formation of motile flagella in *Leishmania*[33]. Recent work using *Trypanosoma brucei* also identified an interaction between Arl3 and Oda16, where the latter is proposed to act as an effector for the ciliary transport of motility-related cargos such as the central pair component HYDIN[34]. Taken together, our work and these other findings indicate that there is a complex interplay between ARL3 and transport factors such as DNAAF9/Shulin and Oda16 during motile cilia formation. Further work is needed to dissect the emerging links between ARL3 and its effector proteins in the trafficking of motility-related cargoes during motile cilia formation.

The sheer scale of axonemal dynein biosynthesis and transport in multiciliated cells could necessitate multiple factors functioning over several steps to concentrate ODAs around and inside growing cilia. Crucially, the need to shut off ODA motor activity is key to preventing aberrant off-target interactions along the trafficking route. Although there could be an intrinsic mechanism in the IFT system to enforce ODA inhibition upon binding, we favor a model where the critical inhibitory role of Shulin may be needed to stabilize IFT–ODA interactions for efficient transport into cilia in *Tetrahymena*. Although our observation that pig ODAs adopt a closed conformation in the presence of DNAAF9 could hint at its inhibitory role in mammalian ODAs (similar to Shulin in *Tetrahymena* ODAs), more functional assays using mammalian ODAs are needed to directly test this in the future. Once inside cilia, we propose that Arl3 in its GTP bound conformation promotes the release of Shulin from ODAs and sequesters it to prevent rebinding. The Shulin–Arl3 complex could exit cilia through diffusion or IFT for recycling or degradation (Fig. 6c). Our displacement model is compatible with ODAs undergoing an irreversible structural reconfiguration from a 'closed' (inhibited) to an 'open' (active) state, upon the release of inhibition, specifically inside cilia during their stable incorporation into a growing ciliary axoneme.

## Online content

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

## Methods

The presented research complied with all relevant ethical regulations. All procedures on mice at The Center for Phenogenomics (TCP) were reviewed and approved by TCP's Animal Care Committee. TCP is certified by the Canadian Council on Animal Care and registered under the Animals for Research Act of Ontario. TCP is part of the IMPC. All procedures and experiments on zebrafish embryos were approved by the Singapore National Advisory on Laboratory Animal Research.

### Immunostaining and image analysis

HAEC cultures were purchased from Epithelix Sarl on day 19 after airlift (early to mid) and on day >45 after airlift (mid to mature). Cells were spread on superfrost slides, air-dried and processed for immunofluorescence. HAECs were fixed for 15 min with 4% PFA and then permeabilized with 0.25% PBS with Triton X-100 for 10-min permeabilization. Cells were blocked in 10% donkey serum in 0.25% PBS with Triton X-100 for 1 h at room temperature and incubated with primary antibodies to DNAAF9 (Proteintech, 23184-1-AP) and DNAI2 (H00064446-M01, clone IC8) in 0.25% PBS with Triton X-100 and 1% donkey serum for 1 h at room temperature or at 4 °C overnight. Cells were incubated with secondary antibodies (Alexa Fluor 568 anti-rabbit antibody (Invitrogen, A-11011) and Alexa Fluor 488 anti-mouse antibody (Invitrogen, A-10680)) for 1 h following three rounds of 15-min 0.25% PBS with Triton X-100 washes. After three final 15-min washes in 0.25% PBS with Triton X-100, cells were counterstained and mounted under a 1.5-mm glass coverslip using prolong gold antifade medium with DAPI. Corners of the coverslip were sealed with commercial nail varnish. All steps were performed at room temperature.

Cells were imaged with a Leica SP8 confocal laser scanning microscope with a ×63 HC PL APO CS2 immersion oil objective. Fluorophores were excited at 488 nm and 568 nm using a 50-mW 405-nm diode and 65-mW argon laser. Lasers were set at 2–5% intensity with an acquisition pinhole of 1. The number of pixels per image was set by first imaging the larger, more elongated late-stage to mature-stage cells and used for smaller and rounder, less elongated early-stage to mid-stage cells. All images were acquired at room temperature using the same parameters for image analysis and quantification.

Image analysis was performed using FIJI ImageJ[35]. Ciliated cells that had no overlapping regions with other cells were included in the analysis. Cilia length measurements were performed by first defining cilia regions as the structure spanning from the first visible apical DNAI2 signal to the ciliary tips. Splayed cilia that appeared detached from the cell body were excluded. Using the ImageJ measuring tool, three distinct cilia regions were measured across the apical length to obtain the average cilia length. Signal intensity measurements were performed by demarcating cilia only, cell only and background regions using the Freehand region of interest (ROI) tool to obtain cilia, cell and background ROIs. Measurements were set to analyze area, integrated density and mean gray values, which were used to calculate the corrected total cilia fluorescence (CTCF), where CTCF = integrated density − (cilia area × mean background fluorescence). The mean background fluorescence used above was calculated by obtaining background intensity with cell bodies included in the ROI without the cilia ROIs and then subtracting the background intensity excluding the cell body and the cilia ROIs. CTCF values were $\log_2$-transformed and plotted using GraphPad Prism software. Unpaired (two-tailed) $t$-test statistics were used to calculate $P$ values.

### Mouse phenotyping data

Mouse gross phenotyping data were obtained from TCP, which is part of the IMPC[36]. Gross phenotyping data on mutant mouse lines (C57BL6/J strain background) are publicly available online (www.mousephenotype.org) or can be accessed upon request.

### In situ hybridization and protein localization studies in zebrafish

For endogenous mRNA expression studies, whole-mount in situ hybridization with zebrafish embryos, using a digoxigenin-labeled antisense probe generated from *dnaaf9* complementary DNA, was performed following a routine procedure. Stained embryos were cleared in glycerol and imaged with a Leica MZ F10 stereomicroscope. For protein localization studies, the coding sequence of zebrafish *dnaaf9* was cloned into either pCS2-Myc vector (5×Myc tag) or pCS2-Xlt vector using the ClonExpress Ultra one-step cloning kit V3 (Vazyme, C117-02). Sense *dnaaf9* mRNA, encoding respective tagged proteins, was synthesized using the mMESSAGE mMACHINE SP6 transcription Kit (Invitrogen, AM1340) with NotI-linearized plasmids as DNA templates. For mRNA injection, 0.5 nl of mRNA at a concentration of 200 ng μl⁻¹ was injected into zebrafish eggs at the one-cell stage, followed by fixation of the fish embryos at specific stages: 10-somite stage, 18-somite stage and 24 h after fertilization with paraformaldehyde-based fixative. Subsequently, the fixed embryos were stained with the following primary antibodies: mouse anti-Myc antibody (Santa Cruz, SC-40) or chicken anti-GFP antibody (Abcam, ab13970) and rabbit anti-acetylated α-tubulin antibody (Cell Signaling, 5335). Alexa anti-mouse 555, anti-rabbit 488 or anti-chicken 488 antibodies were used as secondary antibodies. DAPI was used to counterstain the nuclei. The stained embryos were cleared, mounted in 70% glycerol and imaged using an Olympus FV3000 upright confocal microscope.

### Immunoprecipitation, MS and immunoblotting

Endogenous DNAAF9 immunoprecipitations were performed using HAECs (MucilAir, Epithelix Sarl) at early to mid (19 days after ALI) and mid to mature (45 days after ALI) stages of differentiation. Whole-cell lysates were obtained by incubating cells in IP lysis buffer (50 mM Tris-HCl pH 7.5, 100 mM NaCl, 10% glycerol, 0.5 mM EDTA, 0.5% IGEPAL, one EDTA-free protease inhibitor tablet, 5 μM MG132 and 1 mM DTT). Cells were passed through a 25-gauge needle several times to ensure efficient homogenization and lysis. For preserving potential Hsp90 interactions, 25 mM sodium molybdate (Sigma-Aldrich) was included in the IP lysis buffer to reduce adenosine triphosphate (ATP) hydrolysis and client release. An equivalent concentration of whole-cell lysates was obtained from early–mid-stage and mid–mature-stage cultures (13.7 mg ml⁻¹ and 14.0 mg ml⁻¹, respectively). Lysates from both cell stages were incubated for up to 4 h at 4 °C with an antibody to DNAAF9/C20ORF194 (Proteintech, 23184-1-AP) and an isotype-matched IgG rabbit polyclonal antibody (Proteintech, 30000-0-AP) as a control. Immunocomplexes in the antibody–lysate solutions were concentrated onto Pierce protein A/G magnetic beads (Thermo Fisher Scientific) by incubating for 30 min at 4 °C. Two washes were performed in the IP lysis buffer followed by two washes in the same buffer but with the IGEPAL concentration reduced to 0.2%. A final two washes were performed in the IP lysis buffer without any detergent.

Following washes, beads were frozen at −20 °C for subsequent on-bead tryptic digestion and MS analysis. For immunoblotting analyses, immunocomplexes were eluted by boiling the beads in elution buffer (20% LDS with 10% DTT in Milli Q water) and resolved on NuPAGE Novex 4–12% Bis–Tris protein gels (Life Technologies). Proteins were transferred onto PVDF membranes using the XCell II Blot module (Life Technologies) and blocked in 5% BSA followed by immunoblotting with the following antibodies: anti-DNAAF9 (Proteintech, 23184-1-AP), anti-DNAI2 (M01, clone IC8; Abnova), anti-IFT74 (Proteintech, 27334-1-AP), anti-IFT81 (Proteintech, 11744-1-AP) and anti-Strep tag (GT661, Thermo Fisher Scientific, MA5-17283). A previously validated custom generated antibody for the *Tetrahymena* ODA holocomplex[8] (Eurogentec) was also used to detect ODA subunits by immunoblotting. All primary antibody dilutions were in the range of 1:500–1:1,000 in PBST (0.1% Tween-20) with 1% BSA. Membranes were washed three times in PBST (0.1% Tween-20) and incubated with horseradish peroxidase

(HRP)-conjugated rabbit or mouse secondary antibodies (GeneTex, EasyBlot anti-rabbit IgG and anti-mouse IgG HRP-conjugated mono-clonal antibodies; GTX221666-01-S and GTX221667-01-S, respectively) at 1:1,000 dilution in PBST (0.1% Tween-20) with 1% BSA. Protein bands were detected with SuperSignal West Femto reagent (Thermo Fisher Scientific) and membranes were visualized on an Odyssey Fc Imager (LI-COR Biosciences) or an iBright FL1000 Imaging System under chemiluminescence settings (Thermo Fisher Scientific).

## Proteomics

**TMT labeling and high-pH reverse-phase chromatography.** Immunoprecipitated protein samples were reduced (10 mM TCEP, 55 °C for 1 h), alkylated (18.75 mM iodoacetamide, room temperature for 30 min) and then digested from the beads with trypsin (1.25 µg of trypsin, 37 °C, overnight). The resulting peptides were then labeled with TMTpro 16-plex reagents according to the manufacturer's protocol (Thermo Fisher Scientific) and the labeled samples were pooled and desalted using a SepPak cartridge according to the manufacturer's instructions (Waters). Eluate from the SepPak cartridge was evaporated to dryness and resuspended in buffer A (20 mM ammonium hydroxide pH 10) before fractionation by high-pH reverse-phase chromatography using an Ultimate 3000 liquid chromatography (LC) system (Thermo Fisher Scientific). In brief, the sample was loaded onto an XBridge BEH C18 Column (130 Å, 3.5 µm, 2.1 mm × 150 mm; Waters) in buffer A and peptides were eluted with an increasing gradient of buffer B (20 mM ammonium hydroxide in acetonitrile, pH 10) from 0% to 95% over 60 min. The resulting fractions (six in total) were evaporated to dryness and resuspended in 1% formic acid before analysis by nano-LC–MS/MS using an Orbitrap Fusion Lumos MS instrument (Thermo Fisher Scientific).

**Nano-LC–MS.** High-pH RP fractions were further fractionated using an Ultimate 3000 nano-LC system in line with an Orbitrap Fusion Lumos MS instrument (Thermo Fisher Scientific). In brief, peptides in 1% (v/v) formic acid were injected onto an Acclaim PepMap C18 nano-trap column (Thermo Fisher Scientific). After washing with 0.5% (v/v) acetonitrile and 0.1% (v/v) formic acid, peptides were resolved on an Acclaim PepMap C18 reverse-phase analytical column (250 mm × 75 µm; Thermo Fisher Scientific) over a 150 min organic gradient, using seven gradient segments (1–6% solvent B over 1 min, 6–15% B over 58 min, 15–32% B over 58 min, 32–40% B over 5 min, 40–90% B over 1 min, hold at 90% B for 6 min and reduction to 1% B over 1 min) with a flow rate of 300 nl min⁻¹. Solvent A was 0.1% formic acid and solvent B was aqueous 80% acetonitrile in 0.1% formic acid. Peptides were ionized by nanoelectrospray ionization at 2.0 kV using a stainless-steel emitter with an internal diameter of 30 µm (Thermo Fisher Scientific) and a capillary temperature of 300 °C. All spectra were acquired using an Orbitrap Fusion Lumos MS instrument controlled by Xcalibur 3.0 software (Thermo Fisher Scientific) and operated in data-dependent acquisition mode using a synchronous precursor selection (SPS) MS3 workflow. FTMS1 spectra were collected at a resolution of 120,000, with an automatic gain control (AGC) target of 200,000 and a maximum injection time of 50 ms. Precursors were filtered with an intensity threshold of 5,000, according to charge state (to include charge states 2–7) and with monoisotopic peak determination set to 'peptide'. Previously interrogated precursors were excluded using a dynamic window (60 s, ±10 ppm). The MS2 precursors were isolated with a quadrupole isolation window of 0.7 $m/z$. ITMS2 spectra were collected with an AGC target of 10,000, maximum injection time of 70 ms and collision-induced dissociation collision energy of 35%. For FTMS3 analysis, the Orbitrap was operated at 50,000 resolution with an AGC target of 50,000 and a maximum injection time of 105 ms. Precursors were fragmented by high-energy collision dissociation at a normalized collision energy of 60% to ensure maximal TMT reporter ion yield. SPS was enabled to include up to ten MS2 fragment ions in the FTMS3 scan.

**Data analysis.** The raw data files were processed and quantified using Proteome Discoverer software version 2.4 (Thermo Fisher Scientific) and searched against the UniProt Human database (downloaded January 2023: 81,579 entries) using the SEQUEST HT algorithm. Peptide precursor mass tolerance was set at 10 ppm and MS/MS tolerance was set at 0.6 Da. Search criteria included oxidation of methionine (+15.995 Da), acetylation of the protein N terminus (+42.011 Da) and methionine loss plus acetylation of the protein N terminus (−89.03 Da) as variable modifications and carbamidomethylation of cysteine (+57.0214) and the addition of the TMTpro mass tag (+304.207) to peptide N termini and lysine as fixed modifications. Searches were performed with full tryptic digestion and a maximum of two missed cleavages were allowed. The reverse database search option was enabled and all data were filtered to satisfy a false discovery rate (FDR) of 5%.

**Proteomics data processing.** MS datasets were processed in MS Excel. Protein abundances for each replicate run were used to rank proteins by log₂(fold change) (enrichment) between DNAAF9 IPs over IgG controls. A two-tailed, unpaired $t$-test assuming equal variances was performed to obtain a $P$ value for each protein from triplicate datasets per differentiation stage to rank the hits by significance of enrichment in DNAAF9 IPs versus IgG controls (Source Data). Proteins were filtered against the CRAPome repository set for human proteins (http://www.crapome.org/)[37] using an arbitrary threshold of 50 (that is, proteins appearing in >50 of 716 proteomics experiments using human samples captured in CRAPome). All protein hits with CRAPome values higher than 50 were removed from further analyses except those that had a fold change enrichment of >2.5 (log₂(fold change) of 1.32) and $P > 0.05$ (−log₁₀$P = 1.30$) in the DNAAF9 IPs. These protein hits were included in further analyses and taken forward for validation to account for high abundance 'true' interactors. DNAAF9 interactomes from immature and mature cell stages were visualized as scatter plots (log₂(fold change) versus −log₁₀$P$ value) using GraphPad Prism9.

Three types of data analyses and annotation were performed using the proteomics datasets. First, a comparison of the early-stage and late-stage interactomes was performed to curate a list of proteins that interacted with DNAAF9 at both stages of differentiation. GO analysis on this list of common 87 protein hits was performed using ShinyGO to represent enrichment against a background list of all human protein-coding genes based on cellular component (GO:0005575)[38]. Second, the abundances between the most highly enriched interactors were compared between early and mid stages using ProHits-viz[39] and the fold changes, changes in relative abundances and significance for each protein interactor were visualized. Lastly, unique protein interactors that were found exclusively in the early-stage or late-stage datasets were identified and bioinformatics searches were performed to annotate their cellular localizations.

## Identification of proteins by MS

Two separate types of MS analyses were performed for protein identification. For identifying interactors of the GTP-locked Arl3-Q70L variant in *Tetrahymena*, eluates from strep pulldowns were resolved on an SDS–PAGE gel and silver-stained to visualize bands for coeluting proteins. In parallel, the same eluates were tryptically digested in solution for MS analysis (Source Data). For identifying pig ODA subunits cofractionating together in fraction 14 of the 5–30% sucrose density gradient, Sypro ruby-stained gel slices were excised from the entire lane corresponding to fraction 14 and subjected to in-gel tryptic digestion followed by MS analysis (Source Data).

## Cloning and protein expression in insect cells or bacteria

All genes were codon-optimized for insect cell or bacterial expression. The gene sequence encoding *T. thermophila* Shulin (Q22YU3) was expressed in insect cells using Addgene plasmid 170315 and purified as described previously[8]. *T. thermophila* small GTPase Arl3 (Q229S0)

codon-optimized gene sequences encoding the wild type, Q70L and T30N variants, and human C20ORF194/DNAAF9 (Q5TEA3) were synthesized (Epoch Life Science or Twist Bioscience) and cloned into the pACEBac1 vector containing a C-terminal 2×Strep tag using Gibson assembly for insect cell expression. Human IFT74 (Q96LB3) and IFT81 (Q8WYA0) insect cell expression constructs were obtained from the A.J.R. lab. IFT74 was untagged and coexpressed with His-ZZ-tagged IFT81 for complex formation. The human ARL3 (P36405) Q71L, T31N and triple mutant (F51A;Y77A;Y81A) variants and the *T. thermophila* Arl3 (Q229S0) Q70L, T30N and triple mutant (F50A;Y76A;Y80A) variants were gene synthesized and cloned into a pET22b(+) vector containing a C-terminal His tag for periplasmic bacterial expression (Epoch Life Science).

For insect cell expression, plasmids were first amplified by transforming chemically competent DH5α cells (Thermo Fisher Scientific). To generate baculovirus DNA (bacmid), chemically competent DH10 EmBacY cells (Thermo Fisher Scientific) were transformed and recombinant bacmids were selected by blue–white screening. Purified bacmids carrying the gene of interest were used to transfect *Spodoptera frugiperda*-derived Sf9 cells with FuGENE (Promega) or XtremeGENE (Merck) transfection reagents following the supplier's instructions. Sf9 cells were obtained from the Eukaryotic Expression Facility in the School of Biochemistry at the University of Bristol. On day 4 after transfection, infected Sf9 cells were inspected under a fluorescence microscope for yellow fluorescent protein expression to monitor virus production. Low-titer P1 baculoviruses were collected from the culture suspension and used for virus amplification to produce P2 baculoviruses on day 7 after transfection. For recombinant protein production, 5 ml of P2 baculoviruses were used to inoculate 500 ml of Sf9 cells at a density of $2 \times 10^6$ cells per ml in SF 900 II SFM medium (Gibco). Cells were harvested on day 3 after infection by centrifuging at 2,000$g$ for 15 min at 4 °C using a JLA 8.1000 rotor (Beckman Coulter). Insect cell pellets were frozen in liquid nitrogen for storage or used immediately for protein purification.

For bacterial cell expression, the plasmid pET22b(+) carrying ARL3-FYY was transformed into chemically competent BL21 (DE3) Rosetta 2 pLysS cells (Merck). A saturated overnight preculture of BL21 (DE3) cells (optical density at 600 nm ($OD_{600}$) of 0.05) was used to inoculate 2 l of Luria–Bertani medium (Sigma) supplemented with 100 µg ml$^{-1}$ ampicillin (Sigma) at 37 °C. At an $OD_{600}$ of 0.6–0.8, expression was induced by adding IPTG at a final concentration of 0.5 mM. Upon induction, the temperature was decreased to 18 °C for overnight protein expression. Cells were harvested by centrifuging at 5,000$g$ for 15 min at 4 °C using a JLA 8.1000 rotor (Beckman Coulter). Bacterial cell pellets were frozen in liquid nitrogen for storage or used immediately for protein purification.

## Protein purification

Human C20ORF194/DNAAF9, IFT74 and IFT81 and *T. thermophila* Shulin and Arl3 proteins (wild type, Q70L and T30N) were purified from Sf9 cells as described previously[8]. Briefly, cell pellets were mechanically lysed in 20 ml of lysis buffer (20 mM HEPES–NaOH pH 7.2, 100 mM NaCl, 2 mM magnesium acetate, 1 mM EDTA, 10% (v/v) glycerol and 1 mM DTT) using a Dounce homogenizer (Wheaton) for up to 100 strokes on ice. Lysates were clarified by ultracentrifugation at 47,000$g$ for 45 min at 4 °C in a Ti70 rotor (Beckman Coulter). Clarified lysates were loaded using a sample pump or super-loop (Cytiva) on a 5-ml Strep-Trap HP column (GE Healthcare), which was preequilibrated with five column volumes (CVs) of lysis buffer. The resin was washed with ten CVs of lysis buffer to remove nonspecific binding of contaminants. Recombinant proteins were eluted off the resin by passing five CVs of the elution buffer (lysis buffer containing 3 mM desthiobiotin (IBA)). Eluates were resolved on NuPAGE Novex 4–12% Bis–Tris protein gels (Life Technologies) and stained with Sypro ruby (Invitrogen) to assess the purity of the recombinant proteins. Eluted proteins were further purified over gel filtration at 4 °C on a Superdex 200 10/300 GL column (GE Healthcare) using lysis buffer. Purified proteins eluted in single peaks and were snap-frozen in liquid nitrogen for storage at −70 °C.

His-tagged human ARL3 Q71L, T31N and triple mutant were purified from bacterial pellets. Pellets were resuspended in 35 ml of lysis buffer (50 mM HEPES pH 7.4, 200 mM NaCl, 5% glycerol, 4 mM MgCl$_2$ and 1 mM DTT) supplemented with GDP at 10 µM and lysed on ice by two rounds of sonication for 2 min each with 10-s pulses and 15-s pauses at 50% amplitude using a 435-C probe (Sonics Vibra Cell). Lysates were clarified by centrifugation at 20,000$g$ for 45 min at 4 °C in a JA25.50 rotor (Beckman Coulter). Clarified lysates were applied to a gravity flow column several times over 2 ml of Ni-NTA resin (Sigma), which was preequilibrated with five CVs of lysis buffer. The resin was washed with ten CVs of lysis buffer containing 20 and 40 mM imidazole pH 8 (Sigma) to remove nonspecific binding of contaminants. Recombinant proteins were eluted off the resin in four fractions by passing four CVs of the elution buffer (lysis buffer containing 400 mM imidazole pH 8 (Sigma)). Eluates were resolved on NuPAGE Novex 4–12% Bis–Tris protein gels (Life Technologies) and stained with Instant blue (Novus Biotech) to assess the purity of the recombinant proteins. Eluted proteins were further purified over gel filtration at 4 °C on a Superdex 200 increase 10/300 GL column (Cytiva) using lysis buffer A. Proteins were snap-frozen in liquid nitrogen and stored at −80 °C for further use.

## Purification of three-headed ODA complexes from *Tetrahymena* cilia

Axonemal ODAs were purified using high-salt extraction of cilia as previously described[8]. Briefly, large-scale (6-L) *Tetrahymena* cultures grown in SPP medium containing glucose and FeCl$_3$ were deciliated using dibucaine (0.3 mM). Cell suspensions were centrifuged at 1,500 $g$ for 10 min at 4 °C to pellet cell bodies. Cilia supernatant was pelleted at 4 °C several times with low-speed (1,500$g$) and high-speed (13,500$g$) spins to eliminate mucus and cell body contaminants. Isolated cilia were finally pelleted at 13,500$g$ at 4 °C. Cilia pellets were washed in cilia isolation buffer (CIB; 20 mM HEPES pH 7.4, 100 mM NaCl, 4 mM MgCl$_2$ and 0.1 mM EDTA) and demembranated with 0.25% Triton X-100 detergent, freshly added protease inhibitors, 1 mM DTT and 1 mM PMSF, followed by a 30-min incubation on ice. Following a wash in CIB to remove excess Triton X-100 and another spin to pellet demembranated axonemes, complexes bound to the axonemes were extracted by incubating the axoneme pellet for 30 min on ice in 3 ml of a high-salt buffer (20 mM HEPES pH 7.4, 600 mM NaCl, 4 mM MgCl$_2$, 0.1 mM EDTA, 1 mM DTT, 0.1 mM ATP and 1 mM PMSF). Then, 0.5 ml of the high-salt extract was loaded onto six identical 5–25% sucrose density gradients made in CIB and centrifuged for 16 h at 33,000 rpm (186,200$g$) in an SW41 rotor (Beckman Coulter) at 4 °C. Fractions containing ODA complexes (assessed by SDS–PAGE and gel staining) were pooled and further purified using a Capto HiRes Q 5/50 column (Cytiva). ODA complexes eluting at 300 mM salt were inspected for purity and intactness (assessed by coelution of HCs, ICs and LCs) by SDS–PAGE and Instant blue, Sypro ruby or Lumitein gel staining. ODA-containing peak fractions were flash-frozen in liquid nitrogen in 50% glycerol or used freshly for reconstitution and displacement experiments.

## Purification of two-headed ODAs from pig tracheal cilia and reconstitution with DNAAF9

Pig tracheal ODA complexes were purified following a previously described protocol[40]. Six pig tracheas were freshly acquired from a butcher (S. Leech) directly after slaughter and transported in ice-cold PBS (the pigs were slaughtered as part of the butcher's routine operations). Tracheas were cleaned by removing excess connective tissue and fat. Cleaned tracheas were further washed. Deciliation was performed using buffer A (20 mM Tris pH 7.4, 50 mM NaCl, 10 mM CaCl$_2$, 1 mM EDTA, 7 mM β-mercaptoethanol, 0.1% Triton X-100, 1 mM PMSF, MG132 and four Protease inhibitor tablets). Tracheas were vigorously

shaken in the deciliation buffer for a minute. The buffer was poured into 1-L bottles and centrifuged multiple times to pellet cellular debris at 1,500g for 2 min each time at 4 °C. The clarified supernatant devoid of cellular debris was transferred into 250-ml bottles and centrifuged in in a JLA16.25 rotor (Beckman Coulter) at 12,000g for 5 min at 4 °C to pellet cilia. The cilia pellet was resuspended in 25 ml of resuspension buffer (20 mM Tris pH 7.4, 50 mM KCl, 4 mM MgSO$_4$, 1 mM DTT, 1 mM EDTA, 1 mM PMSF and one protease inhibitor tablet) and centrifuged again to obtain a cleaner cilia pellet. A high-salt extract containing ODAs and other axoneme-bound proteins was obtained by resuspending the cilia pellet in 3 ml of dynein extraction buffer (20 mM Tris pH 7.4, 600 mM KCl, 4 mM MgSO$_4$, 1 mM DTT, 1 mM EDTA, 1 mM PMSF and 0.1 mM ATP) for 30 min on ice. The high-salt extract was centrifuged at 31,000g for 15 min and the supernatant was dialyzed into a lower-salt dialysis buffer (20 mM HEPES, 50 mM NaCl, 4 mM MgCl$_2$, 1 mM DTT, 0.1 mM ATP and 1 mM PMSF) for 2 h. Then, 50 mM NaCl was used in the dialysis buffer to dialyze out KCl, which impairs gradient fractionation. Next, 0.5 ml of the dialysate was overlaid onto a 5–30% sucrose gradient made in dialysis buffer and resolved by centrifuging the gradients overnight (15–16 h) at 168,000g in an SW40 rotor (Beckman Coulter). The gradient was manually fractionated into 0.5-ml fractions and a small sample from each fraction was run on an SDS–PAGE gel and stained with Sypro ruby to visualize protein bands. Fraction 14 contained high-molecular-weight bands that accumulated strong stain. To verify that these high-molecular-weight bands corresponded to ODA HCs, fraction 14 was subjected to MS identification of all proteins cofractionating in fraction 14 (Extended Data Fig. 4 and Source Data). Negative-stain EM analysis was performed on fraction 14 to verify the intactness of the purified ODA holocomplexes. Complexes in fraction 14 were also mixed with DNAAF9 and negatively stained to check the effect of DNAAF9 on ODA conformation compared to free ODA.

## Complex reconstitutions, size-exclusion chromatography and Shulin displacement assay

Complex formations involving *Tetrahymena* Arl3-Q70L, Arl3-T30N, Arl3-Q70L;FYY and Shulin and human ARL3-Q71L, ARL3-T31N, ARL3-Q71L;FYY and DNAAF9 were investigated by analytical size-exclusion chromatography using a Superdex 200 increase 3.2/300 column (Cytiva). For cross-species analysis, complexes between DNAAF9 and Arl3 variants were resolved on a Superdex 200 increase 10/300 column (Cytiva). Shulin or DNAAF9 was incubated with a four-fold molar excess of Arl3 or ARL3 variants (1,880 nmol), respectively, for 30 min on ice. The mix was supplemented with 1 mM GDP or GTP, applied to the size-exclusion chromatography column and eluted with one CV of lysis buffer A. The elution profile was recorded and eluted fractions were analyzed by SDS–PAGE and stained with Instant blue (Novus Biotech) or Lumitein gel stain (Biotium) to assess coelution of the recombinant proteins as an indicator of complex formation between Shulin or DNAAF9 and the various Arl3 or ARL3 variants, respectively.

Displacement assays were performed using native ODAs purified from isolated *Tetrahymena* cilia and recombinant *Tetrahymena* Shulin because of technical limitations in purifying mammalian ODAs. *Tetrahymena* ODAs were reconstituted with Shulin as described previously[8]. Purified fractions containing the reconstituted complex were combined and divided into two equal fractions. Each fraction was incubated with an excess of Arl3-Q70L or Arl3-Q70L;FYY (6.5 μg) in the presence of 1 mM GTP for 30 min on ice. The resulting mixtures were resolved on a Superose 6 increase 3.2/300 column (Cytiva) and the eluted fractions were analyzed by SDS–PAGE and Sypro ruby staining. Band intensities in the peak fraction were quantified by gel densitometry analysis using FIJI to quantify the amount of Shulin coeluting with ODAs in the presence of Arl3-Q70L or Arl3-Q70L;FYY as a control as a proxy for displacement. A ratio of Shulin to ODA HCs and Shulin to the IC2 IC was calculated on the basis of band intensities in each of the three experimental conditions

performed in triplicate. An analysis of variance (ANOVA) with Tukey's test for multiple comparisons was used as a statistical test to compare mean ratios between the different experimental conditions and the data were plotted using GraphPad Prism9.

Separately purified ODA–Shulin and IFT81–IFT74 complexes were also mixed and resolved on a Superose 6 increase 10/300 column (Cytiva) to test for binding. Peak fractions were subjected to immunoblotting with a custom *Tetrahymena* ODA antibody and antibodies to the Strep tag, IFT74 and IFT81.

## Negative-stain EM and data processing

Freshly isolated complexes between Shulin or DNAAF9 and Arl3-Q70L were diluted to between 0.01 and 0.05 mg ml$^{-1}$ in gel-filtration buffer and 3 μl of each sample was applied to freshly glow-discharged grids (300-mesh copper with formvar and carbon support, TAAB). Grids were glow-discharged using an Edwards Sputter Coater S150B or an EMS GloQube system for 15–30 s at 35 mA. Samples were incubated on grids at room temperature for 1 min and then blotted by wicking away excess liquid using a filter paper. Next, 2% uranyl acetate was applied to grids for 1 min and air-dried after wicking away excess liquid. Micrographs were acquired using FEI EPU on a FEI 200-kV Tecnai 20 microscope equipped with an FEI Ceta 4,000 × 4,000 charge-coupled device camera or FEI Flacon II direct electron detector at a nominal magnification of ×80,000 and a pixel size of 1.27 Å per pixel (DNAAF9–Arl3 complex) or 3.64 Å per pixel (Shulin–Arl3 complex).

Samples containing pig ODA holocomplexes alone or combined with DNAAF9 were negatively stained for EM analyses as described above. Micrographs were acquired at a nominal magnification of ×30,000 and a pixel size of 3.803 Å per pixel on a JEOL 2100 Plus 200-kV transmission EM instrument equipped with a Gatan OneView camera. Small datasets of extracted intact particles were used to quantify conformational changes on the basis of 2D classifications and to obtain representative 2D class averages of pig ODAs alone (total intact particles: 3,705; open: 3,594 (97%); closed: 111 (3%)) and pig ODAs in the presence of DNAAF9 (total intact particles: 1,896; open: 908 (48%); closed: 988 (52%)).

All EM data processing was performed using RELION 3.2 or 4. For obtaining the DNAAF9–Arl3 2D class averages and 3D map, 1,232 particles were first manually picked from seven micrographs. An initial 2D classification was performed to obtain class averages that were used to autopick 161,220 particles. Successive rounds of 2D classifications provided well-defined 2D class averages from 21,423 particles. Subclassification of these particles yielded classes that contained DNAAF9 alone (13,444 particles) and DNAAF9 with an extra density corresponding to Arl3 (11,135 particles). The Shulin–Arl3 dataset was similarly processed to obtained 2D class averages of Shulin alone and Shulin bound by Arl3. The 11,135 particles from the DNAAF9–Arl3 dataset were used for 3D classification and refinement to obtain a low-resolution (~18 Å) map of the DNAAF9–Arl3 complex.

## AF2 and AF3 monomer and multimer predictions and structural analyses

AF2 multimer and monomer structural predictions were generated using the Google Colab AF2 Notebook with a Colab Pro+ subscription[41,42]. For multimer prediction, the algorithm was asked to model a 1:1 heteromeric complex between human DNAAF9 (Q5TEA3) and ARL3 (P36405). Default parameters were used for multiple-sequence alignment generation and the Amber relaxation step was enabled. Monomer protein models for DNAAF9 and Shulin (Q22YU3) in Extended Data Fig. 1b,d were also generated using the Google Colab AF2 Notebook. In each prediction, the highest-ranked model was downloaded in .pdb format for further analyses. Model statistics are as generated by the software with annotations to highlight relevant details. AF3 (ref. 19) (AF server: https://alphafoldserver.com/) was used for modeling larger complexes and the presence of ligands.

For predicting site 1 contact, a sequence comprising DNAH5's helical bundles 5–7 was used with the full-length DNAAF9 sequence. For predicting site 2 contacts, a sequence containing DNAH5's motor domain was used with the sequence for DNAAF9's C1–C3 domains. Full-length DNAAF9 and ARL3 sequences were used to predict an interaction between the two in the presence of one molecule of GTP or GDP using the default parameters[19]. Domain-level root-mean-square deviation scores were obtained from the PDB using the tool for pairwise structural alignment[43] using the domain boundaries for Shulin and DNAAF9 as described previously[8], followed by confirmation of structural alignments in UCSF Chimera X[44]. The DNAAF9–ARL3 AF2-predicted model was rigid-body fitted into the negative-stain EM 3D reconstruction of the DNAAF9–Arl3 complex using UCSF ChimeraX.

### Bioinformatics and conservation analysis

Ortholog searches for human DNAAF9 and ARL3 were performed using PSI-BLAST. For DNAAF9, sequences from six representative model organisms with motile cilia (pig, mouse, *Xenopus*, zebrafish, *Tetrahymena* and *Chlamydomonas*) were aligned using the tools embedded in UniProt (Clustal Omega)[45]. Sequence alignments were visualized using ESPript[46]. Residue-level conservation scores were calculated and plotted onto PDB structures using the CONSURF server for the predicted DNAAF9 monomer and the DNAAF9–ARL3 multimer complex[47]. Final visualizations were performed in ChimeraX.

### Statistics and reproducibility

HAECs were obtained as transwell cultures containing separate pooled batches of cells from Epithelix Sarl at two differentiation time points. Cell clumps were gently scraped from transwell inserts and suspended down to single cells, which were then applied to glass slides and allowed to air-dry. Each slide contained several hundred individual cells that were processed for immunostainings. Many cells were observed under a confocal microscope. Of these, 10–20 representative cell images per batch were randomly acquired for quantitative analyses. No statistical method was used to predetermine sample size but our sample sizes are similar to those reported in previous publications[8]. Data distribution was assumed to be normal but this was not formally tested. No data were excluded from the cell immunostaining analyses. Three authors (B.B., M.R. and G.R.M.) were involved in acquiring cell images independently. It was difficult to perform blinding as the cell slides were labeled. The investigators were not blinded to allocation during experiments and outcome assessment. Additional blinding was not attempted. Immunostainings performed on three replicate batches showed similar subcellular distributions for endogenous DNAAF9 and DNAI2 in the immature and more mature human airway cells.

Immunoprecipitation MS experiments were performed in triplicate followed by 12-plex TMT MS runs. ODA–Shulin biochemical reconstitutions and displacement experiments using Arl3-Q70L or Arl3-Q70L;FYY were performed in triplicate and these replicate data were used for gel-densitometry-based quantification analyses.

All zebrafish localization experiments were performed in at least two biological replicates and the numbers of control and injected embryos imaged were >4. Imaging of >4 control and injected embryos from two biological replicates showed similar subcellular distribution for the C-terminal tagged versions of Dnaaf9 protein. For mouse studies, eight male and eight female adult mutant animals from the same C57BL6/J strain background were phenotyped. Brain histopathology was performed on two randomly selected male and two randomly selected female homozygote animals.

Negative-stain EM data were collected until a sufficiently high resolution to answer our biological question was achieved. Sample sizes were not predetermined for the structural analyses performed in this study. No attempt was made to replicate the negative-stain EM data, which involves averaging of several thousands of individual protein molecules. EM micrographs that contained empty areas and thick stains in the field of view were excluded. During 2D and 3D classification, particles that did not yield detailed images to reveal structural densities were also excluded.

### Reporting summary

Further information on research design is available in the Nature Portfolio Reporting Summary linked to this article.

## Data availability

MS data were deposited to ProteomeXchange and made available through the PRIDE[48] partner repository with the primary accession code PXD052722. Additional data are available from figshare (https://doi.org/10.6084/m9.figshare.27195438)[49]. All other data supporting the findings of this study are available from the corresponding author upon reasonable request. Source data are provided with this paper.

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

## Acknowledgements

We thank K. Mochizuki for providing *Tetrahymena* strains, I. Collinson, E. Fodor and G. Isom for sharing critical equipment, and the University of Bristol's Proteomics and Bioimaging Facility (J. Mantell) for data acquisition. We thank A. Flenniken and M. Eskandarian at TCP (IMPC) for providing us with phenotyping and histopathology data on DNAAF9 mutant mice, E. Robertson for help with critical interpretation of these data, and Q. Tao for assistance generating zebrafish in situ hybridization data. We thank S. Murphy, S. Sanyal and

K. Toropova for critical reading of the paper and C. Lau for helpful comments. We thank R. Hughes, K. Ingram, I. Dowell and M. Hassan for assistance acquiring preliminary data and J. Daly for advice on bioinformatic analyses of proteomics data. This work is supported by funds from the Agency for Science, Technology and Research of Singapore (SC15-R0010) to S.R., a Wellcome Senior Research Fellowship (217186/Z/19/Z) to A.J.R., and a Medical Research Council Career Development Award (MR/X007219/1), an Academy of Medical Sciences Springboard Award (SBF007\100151) and startup support from the School of Biochemistry, University of Bristol and the Dunn School of Pathology, University of Oxford (EPA fund) to G.R.M.

## Author contributions

K.H.B.I. purified all the recombinant proteins, conducted the biochemical reconstitutions and displacement experiments and, with G.R.M., performed the bioinformatics, structural modeling and analyses. G.R.M. performed the cellular studies and endogenous immunoprecipitation for proteomics, conducted the initial biochemical experiments, purified the endogenous pig ODAs and, with K.H.B.I., purified the *Tetrahymena* ODAs. G.R.M. prepared the negative-stain EM samples, acquired the micrographs with C.M. and processed all the EM data. With guidance from G.R.M., B.B. analyzed the cellular imaging data and M.R. performed the endogenous IP western blots. H.L. and S.R. conducted the protein tagging and imaging experiments in zebrafish. K.H. performed the TMT labeling and acquired the proteomics datasets, which G.R.M. analyzed and interpreted. A.J.R. provided the expression constructs for the production of recombinant IFT74 and IFT81 proteins. G.R.M. conceptualized the project with K.H.B.I. and guided the project. K.H.B.I. and G.R.M. prepared the figures and wrote the paper with input from all the authors.

## Competing interests

The authors declare no competing interests.

## Additional information

**Extended data** is available for this paper at https://doi.org/10.1038/s41594-025-01680-9.

**Correspondence and requests for materials** should be addressed to Girish R. Mali.

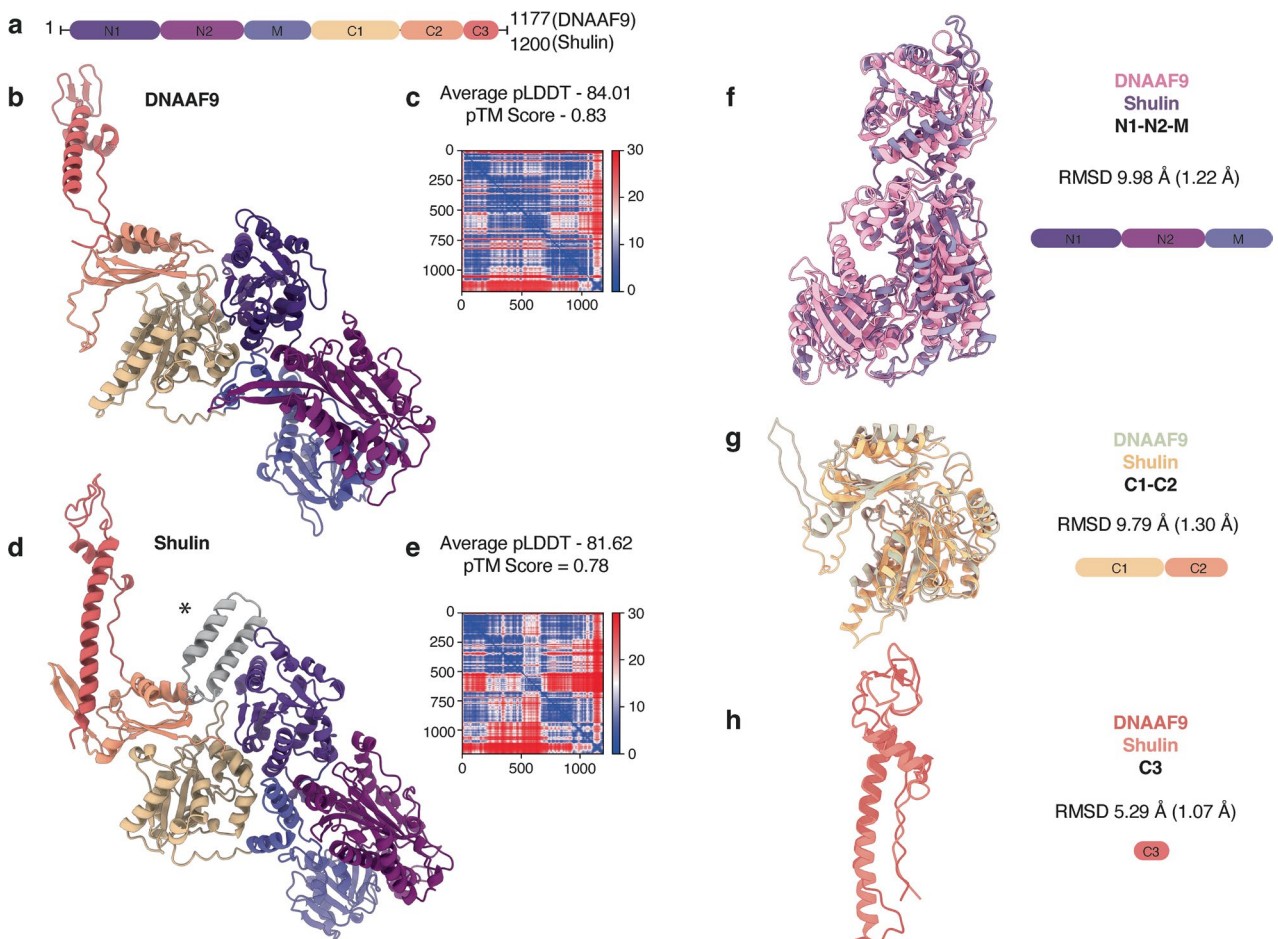

**Extended Data Fig. 1 | DNAAF9 and Shulin share high overall and domain level structural similarity. a**. Domain organisation of DNAAF9 and Shulin. **b, c, d, e**. AlphaFold2 generated models with PAE plots, pLDDT and pTM scores for DNAAF9 and Shulin. **f, g, h**. Structural comparison of DNAAF9 and Shulin divided into sub-domains (f: N1-N2-M domains, g: C1-C2 domains, h: C3 extension) by RMSD analysis. Overall RMSD values are shown with scores for best aligned regions included in brackets. Flexible loop regions were retained for the analysis. * An extra helix-loop-helix (gray) is predicted in Shulin's structure in (**d**); this was excluded for calculating domain level RMSD scores.

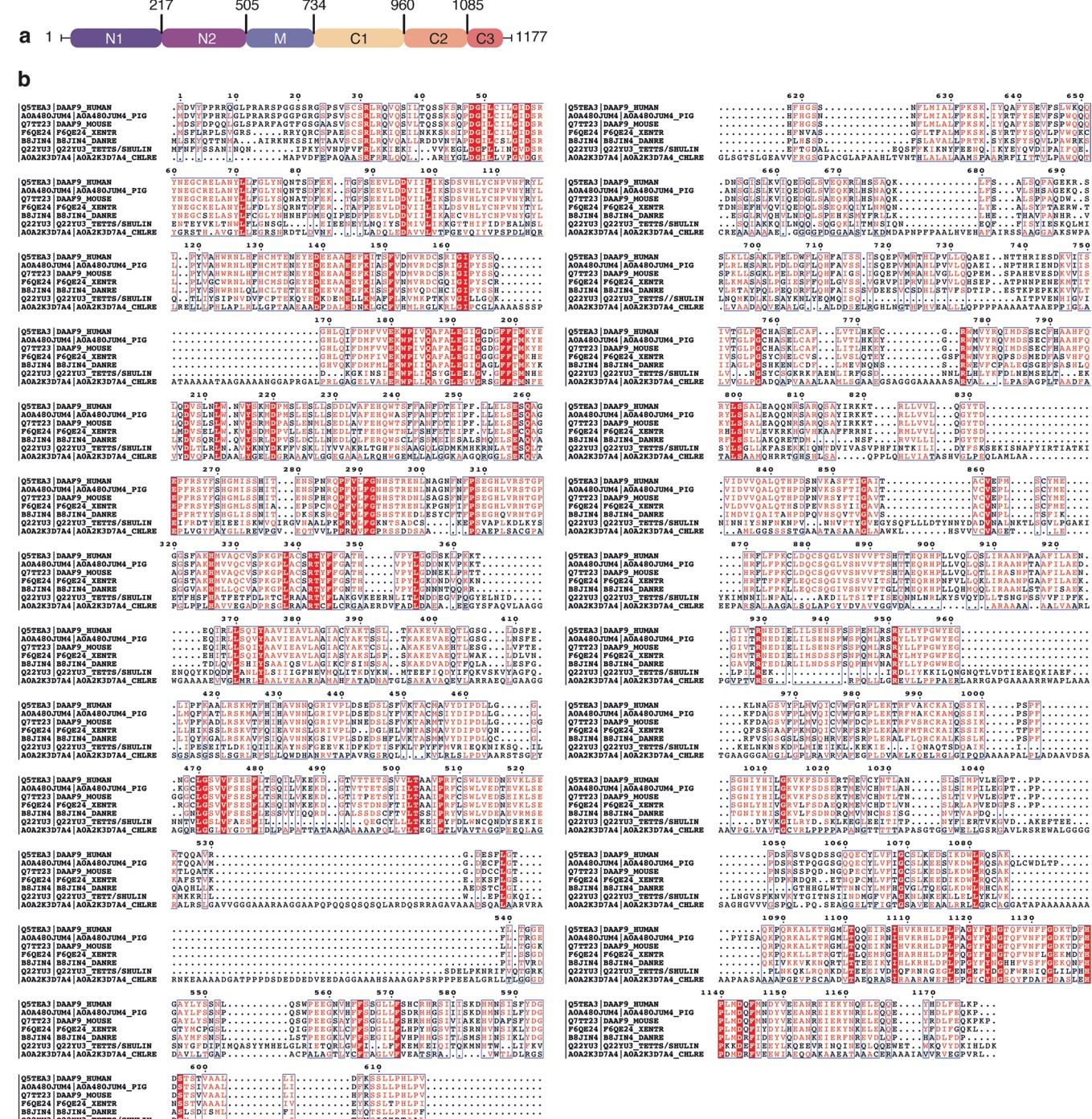

**Extended Data Fig. 2 | DNAAF9 domain architecture and multiple sequence alignment with its orthologs. a**. Domain boundaries of DNAAF9 **b**. Sequence alignment between DNAAF9 and its orthologs across model ciliate species generated using ESPript. UniProt IDs are labelled to the left (Q5TEA3: Human DNAAF9/Shulin; A0A480JUM4: Pig DNAAF9/Shulin; Q7TT23: Mouse DNAAF9/ Shulin; F6QE24: *Xenopus* DNAAF9/Shulin; B8JIN4: Zebrafish DNAAF9/Shulin; Q22YU3: *Tetrahymena* Shulin; A0A2K3D7A4: *Chlamydomonas* DNAAF9/Shulin). Conserved residues are highlighted in red and similar residues are colored in red with blue boxes around.

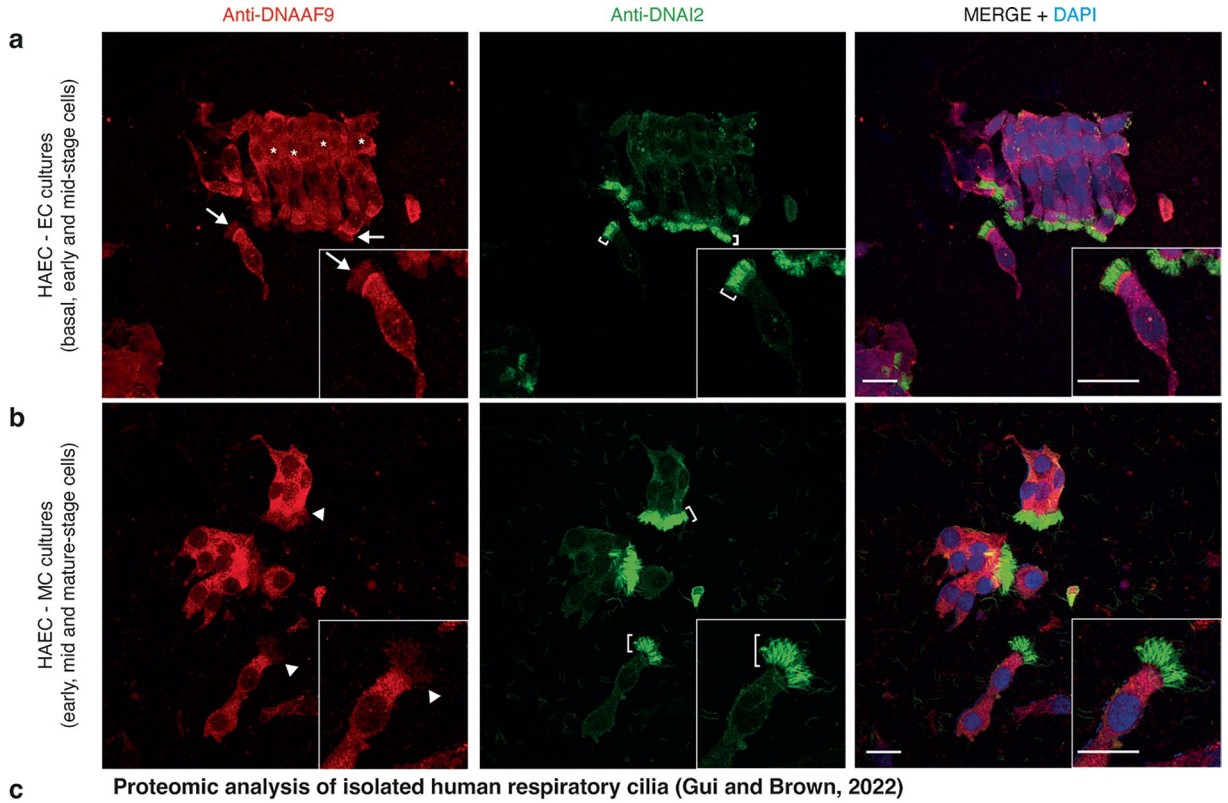

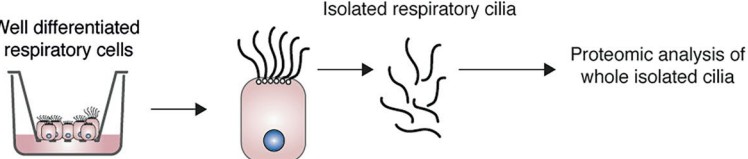

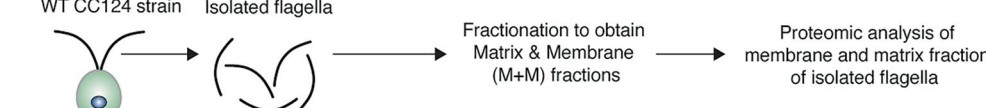

### c. Proteomic analysis of isolated human respiratory cilia (Gui and Brown, 2022)

| Uniprot Reference ID | Gene Symbol | Description | MW kDa | Unique Peptide Count |
|---|---|---|---|---|
| sp\|Q8TE73\|DYH5_HUMAN | DNAH5 | outer arm dynein heavy chain gamma | 528 | 81 |
| sp\|Q9NYC9\|DYH9_HUMAN | DNAH9 | outer arm dynein heavy chain beta | 511 | 52 |
| sp\|Q96DT5\|DYH11_HUMAN | DNAH11 | outer arm dynein heavy chain beta | 520 | 32 |
| sp\|Q9UI46\|DNAI1_HUMAN | DNAI1 | outer dynein arm intermediate chain 1 | 79 | 10 |
| sp\|Q9GZS0\|DNAI2_HUMAN | DNAI2 | outer dynein arm intermediate chain 2 | 68 | 5 |
| **sp\|Q5TEA3\|CT194_HUMAN** | **C20orf194** | **Human DNAAF9 (Shulin ortholog)** | **132** | **2** |

### d. Proteomic analysis of isolated *Chlamydomonas* flagella (Sakato-Antoku and King, 2022)

| Accession Number | Alternate ID | Phytozome 12 Description | MW kDa | WT (CC124) Flagella Average Normalised Total Spectra |
|---|---|---|---|---|
| Cre03.g145127.t1.1 | DHC13, ODA11 | Flagellar outer arm dynein heavy chain alpha | 504 | 69.63 |
| Cre09.g403800.t1.2 | DHC14, ODA4 | Flagellar outer arm dynein heavy chain beta | 520 | 60.70 |
| Cre11.g476050.t1.2 | DHC15, ODA2 | Flagellar outer arm dynein heavy chain gamma | 339 | 58.43 |
| Cre12.g536550.t1.2 | ODA-IC1, IC78 | Flagellar outer dynein arm intermediate chain 1 | 76 | 1.85 |
| Cre12.g506000.t1.2 | ODA-IC2, IC69 | Flagellar outer dynein arm intermediate chain 2 | 64 | 8.20 |
| **Cre11.g467556.t1.1** | **DNAAF9** | ***Chlamydomonas* DNAAF9 (Shulin ortholog)** | **145** | **2.25** |

**Extended Data Fig. 3 | DNAAF9 shows a stage-specific ciliary localisation in HAECs with low abundance in mature isolated cilia. a, b.** Co-immunostaining of DNAAF9 and DNAI2 in ECs with short cilia (arrow and short brackets) and MCs with long cilia (arrowhead and long brackets). Asterisks * in (**a**) show non-multiciliated basal cells in ECs. Inset panels show an enlarged image of an EC and MC for comparison. Representative images of cells from atleast three replicate staining's are shown Scale bars, 20µm **c. d** Analysis of published proteomic datasets containing unique peptides per protein detected by mass spectrometry in isolated mature human airway cilia and normalised spectra per protein detected in the matrix and membrane (M + M) fractions of isolated full-length *Chlamydomonas* flagella from vegetative cells of the CC124 wild-type strain. Unique peptide counts and normalised total spectral values for the listed proteins are shown.

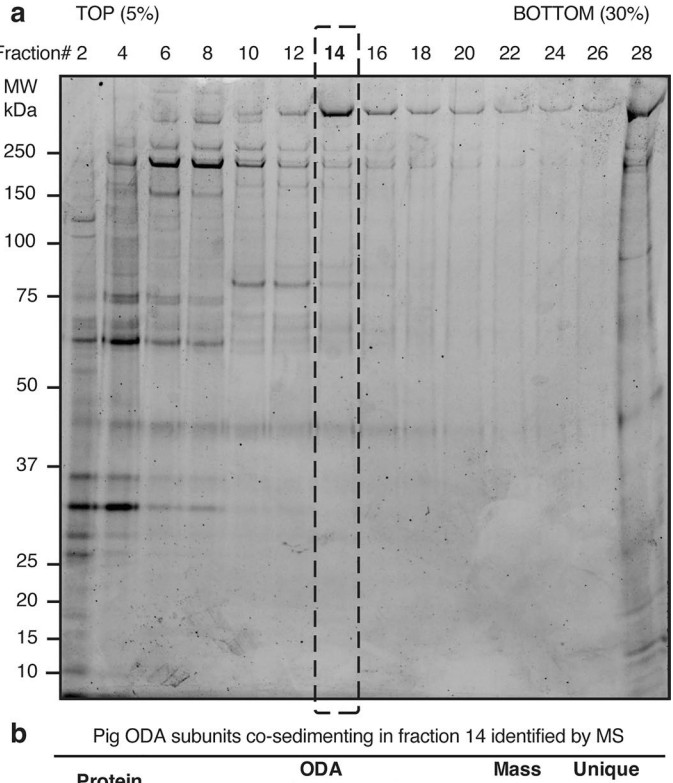

**a**

**b** Pig ODA subunits co-sedimenting in fraction 14 identified by MS

| Protein | ODA Subunit | Mass kDa | Unique Peptides |
|---|---|---|---|
| DNAH9 | Beta Heavy Chain | 509 | 68 |
| DNAH5 | Gamma Heavy Chain | 528 | 66 |
| DNAH11 | Beta Heavy Chain | 517 | 21 |
| DNAI2 | IC2 Intermediate Chain | 66 | 6 |
| DNAL1 | LC1 Light Chain | 22 | 4 |
| NME9 | TXNDC6/LC3 Light Chain | 39 | 4 |

**Extended Data Fig. 4 | DNAAF9 does not co-purify with ODAs extracted from pig tracheal axonemes. a**. SDS-PAGE analysis of fractions from a sucrose density gradient centrifugation resolving complexes obtained from high-salt extraction of pig tracheal cilia axonemes. Fraction 14 shows a major high molecular weight band consistent with the mass of a dynein heavy chain. Molecular weight markers (MW) are shown in kDa. Pig ODA purifications were performed twice with similar results **b**. Mass spectrometry analysis of ODA subunits co-sedimenting in fraction 14. Unique peptide counts detected at a 95% peptide threshold are shown.

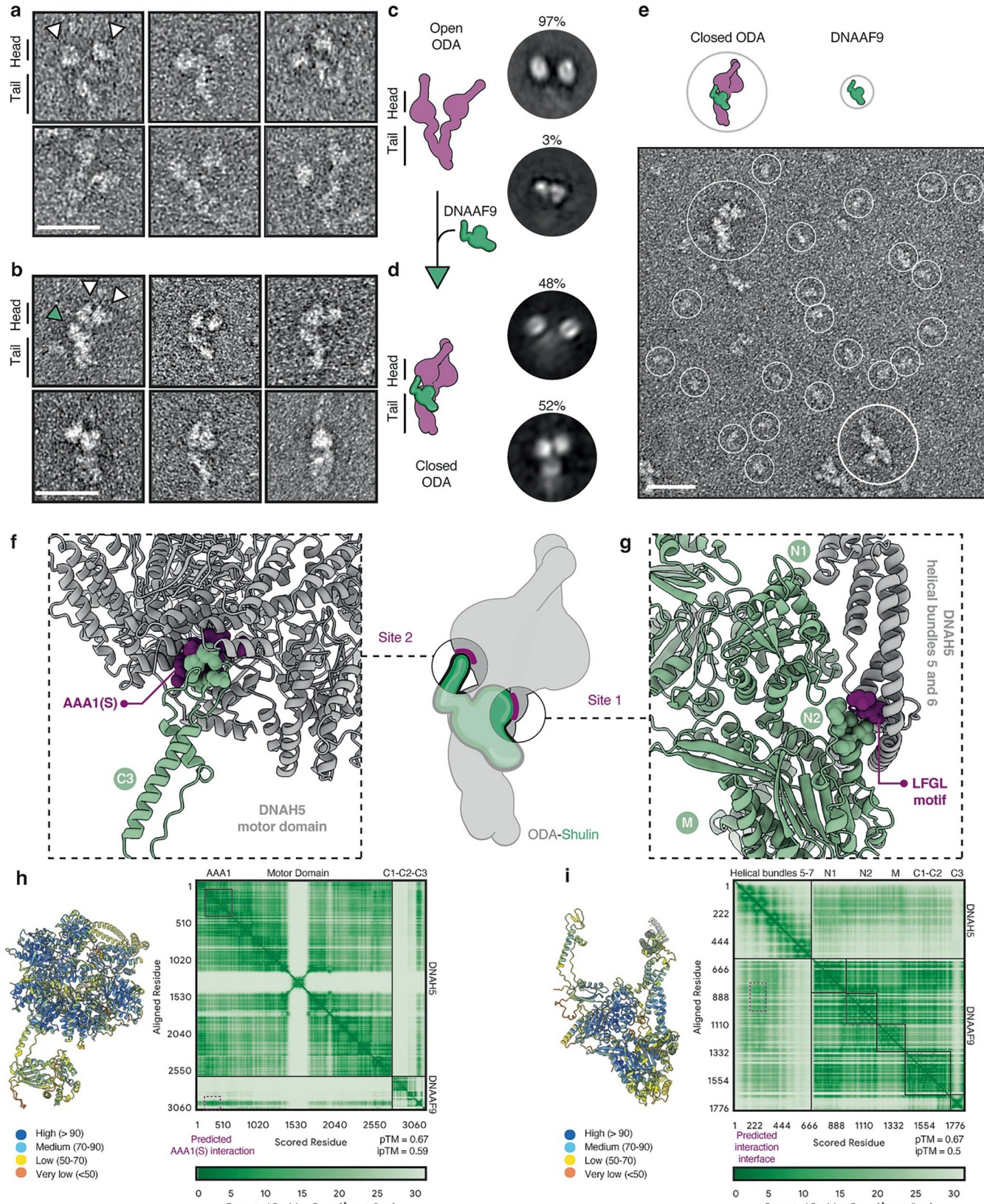

**Extended Data Fig. 5 | See next page for caption.**

**Extended Data Fig. 5 | DNAAF9 binds mammalian ODAs to stabilise a closed conformation. a, b**. Electron micrographs of negatively stained two-headed pig ODA particles in isolation and when combined with recombinant DNAAF9 are shown (n = 2 replicates). Particles adopting open conformation are shown in (**a**) and those with a closed phi-particle like conformation are in (**b**) with white arrowheads marking the positions of the motor heads in top left panels; the green arrowhead in top left panel in (**b**) points to a putative density for DNAAF9. Scale bars, 50 nm. **c, d**. Conformational differences between the two samples are illustrated by a schematic diagram. Representative 2D class averages highlighting the open and closed states are shown to the right with corresponding percentages of open versus closed ODA particles in samples with ODA alone (**c**) and when ODAs were mixed with recombinant DNAAF9 (**d**).

**e**. A representative negative stain electron micrograph with single particles of pig ODA holocomplexes and DNAAF9 encircled. Scale bar, 25 nm. **f, g**. AlphaFold3-Multimer model of key binding interfaces (site 1 and 2) between DNAAF9 and DNAH5 (gamma heavy chain of human ODAs). Conserved interface residues at the AAA1 site in the motor domain and the LFGL motif in the tail helical bundle of DNAH5 are surface represented as purple spheres with the corresponding contacting residues in DNAAF9 shown as green spheres. **h, i**. pLDDT scores are plotted onto the predicted structures to highlight local confidence in the modeling (dark blue = pLDDT 90-100; high confidence; orange = pLDDT 0-50; low confidence). PAE plots highlighting predicted interaction interfaces and model statistics are shown.

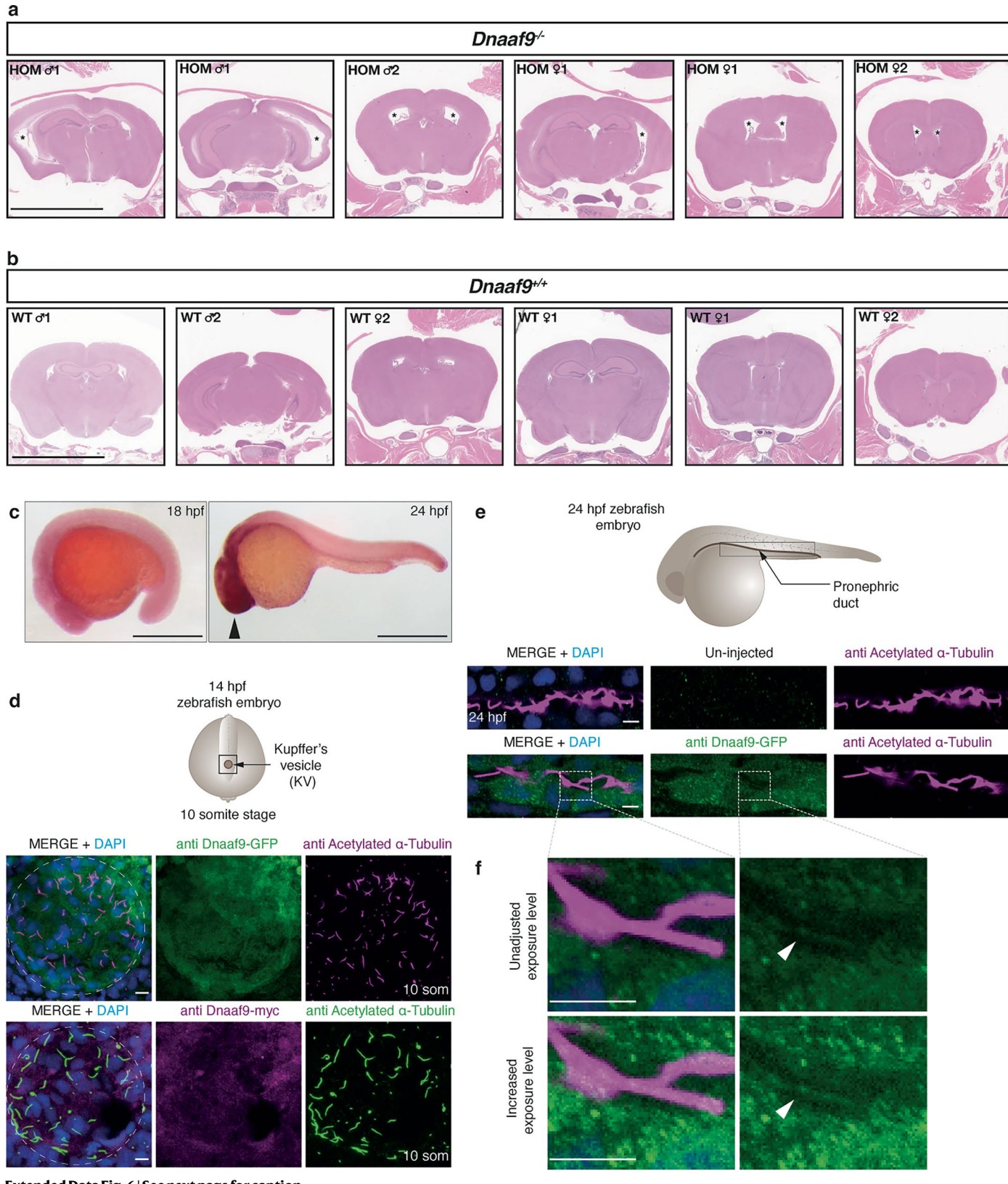

**Extended Data Fig. 6 | See next page for caption.**

**Extended Data Fig. 6 | *Dnaaf9*⁻/⁻ mutant mice develop mild hydrocephalus and in zebrafish it is expressed broadly with a predominantly cytoplasmic localization in motile ciliated cells. a, b**. Histopathology images of brain coronal sections obtained from 4 homozygote (2 males and 2 females) animals at 16 weeks from a *Dnaaf9*⁻/⁻ line and 4 wildtype (2 males and 2 females) control animals. Both homozygous and wildtype animals were bred on a C57BL6/J background strain. Asterisks * in the panel of images in (a) denote dilatation of the lateral ventricles in all four *Dnaaf9*⁻/⁻ animals indicative of hydrocephalus. **c**. Endogenous *dnaaf9* mRNA expression analysis using *in situ* hybridization on zebrafish embryos at 18- and 24-hours post-fertilization (hpf) showing broad expression with enrichment in cranial regions (black arrowhead). **d**. Cartoon showing position of the Kupffer's vesicle (KV) in 14-hour post-fertilization (14 hpf) zebrafish embryos at the 10-somite stage (10 som) and co-immunostaining panels showing Dnaaf9-GFP or Dnaaf9-myc, acetylated α-tubulin to mark motile monocilia in the KV and merge with DAPI. The KV cells are encircled with dashed circles. Dnaaf9-GFP (green) and Dnaaf9-Myc (magenta) localized to the cytoplasm of motile monocilia-bearing KV cells of injected embryos. **e. f**. Cartoon showing position of the pronephric duct (kidney duct) in a 24-hour post-fertilization (24 hpf) zebrafish embryo. Motile cilia in the pronephric duct of a control un-injected embryo and an embryo injected with Dnaaf9-GFP (green), stained with acetylated tubulin antibodies (magenta) and merge with DAPI are shown. Dnaaf9-GFP (green) localized predominantly to the cytoplasm of motile cilia-bearing pronephric duct cells of an injected embryo. Occasional weak ciliary signal for Dnaaf9-GFP detected in some cells (white arrowhead) is shown unadjusted and at a higher exposure level. Scale bars in (**a**, **b**) are 5 mm, in (**c**) are 500 μm and in (**d**-**f**) are 5 μm.

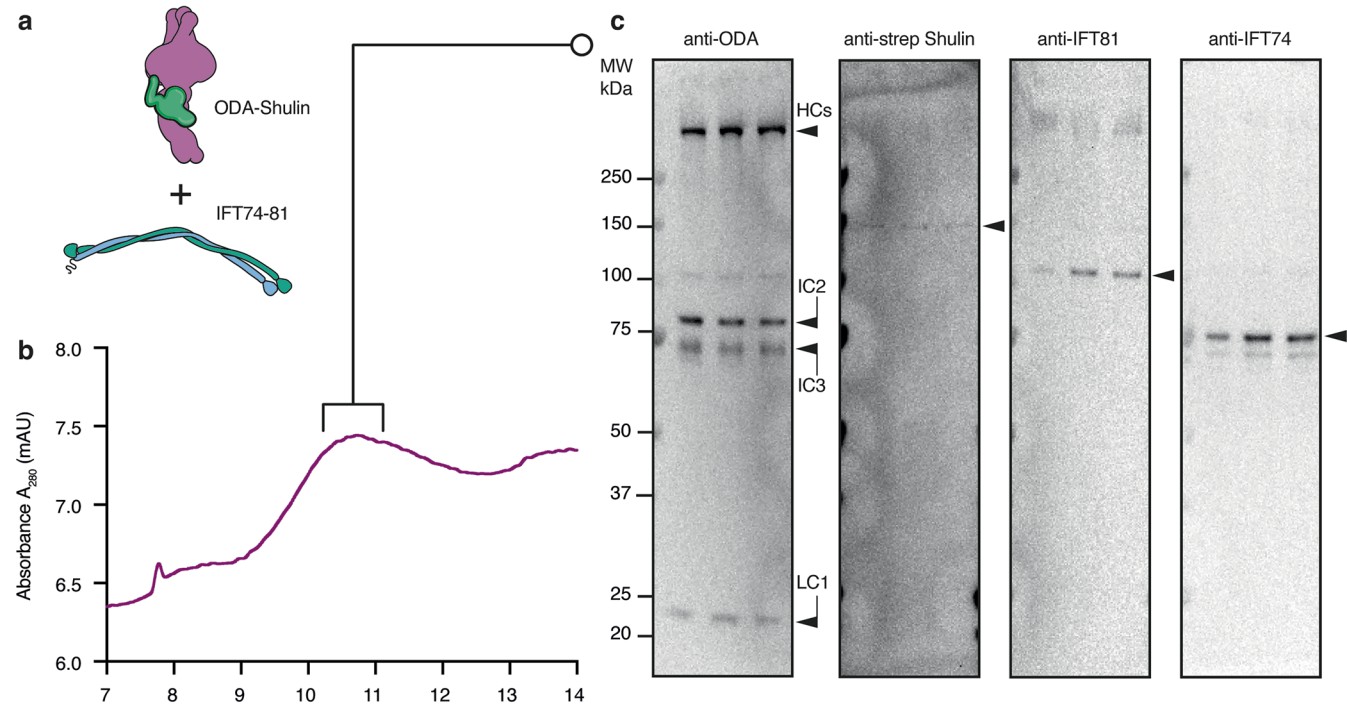

**Extended Data Fig. 7 | Coupling of Shulin bound ODAs to the IFT system can occur via the IFT74/81 dimer. a**. Schematic showing reconstitution between the purified ODA-Shulin complex and the IFT74/81 heterodimeric complex. **b**. Analytical size exclusion trace of the reconstituted ODA-Shulin and IFT74/81 complexes with the main peak shown. **c**. Peak fractions resolved on an SDS-PAGE gel and subjected to immunoblot analysis are shown. Fractions were probed with specific antibodies to detect the *Tetrahymena* ODA holocomplex, strep-tagged recombinant Shulin, His-ZZ tagged recombinant human IFT81 and untagged human IFT74 which forms an obligate complex with IFT81. Bands detecting the various components co-eluting in the peak fractions are highlighted indicating the formation of a ternary complex. Reconstitutions were performed twice with similar results. Molecular weight markers (MW) are shown in kDa.

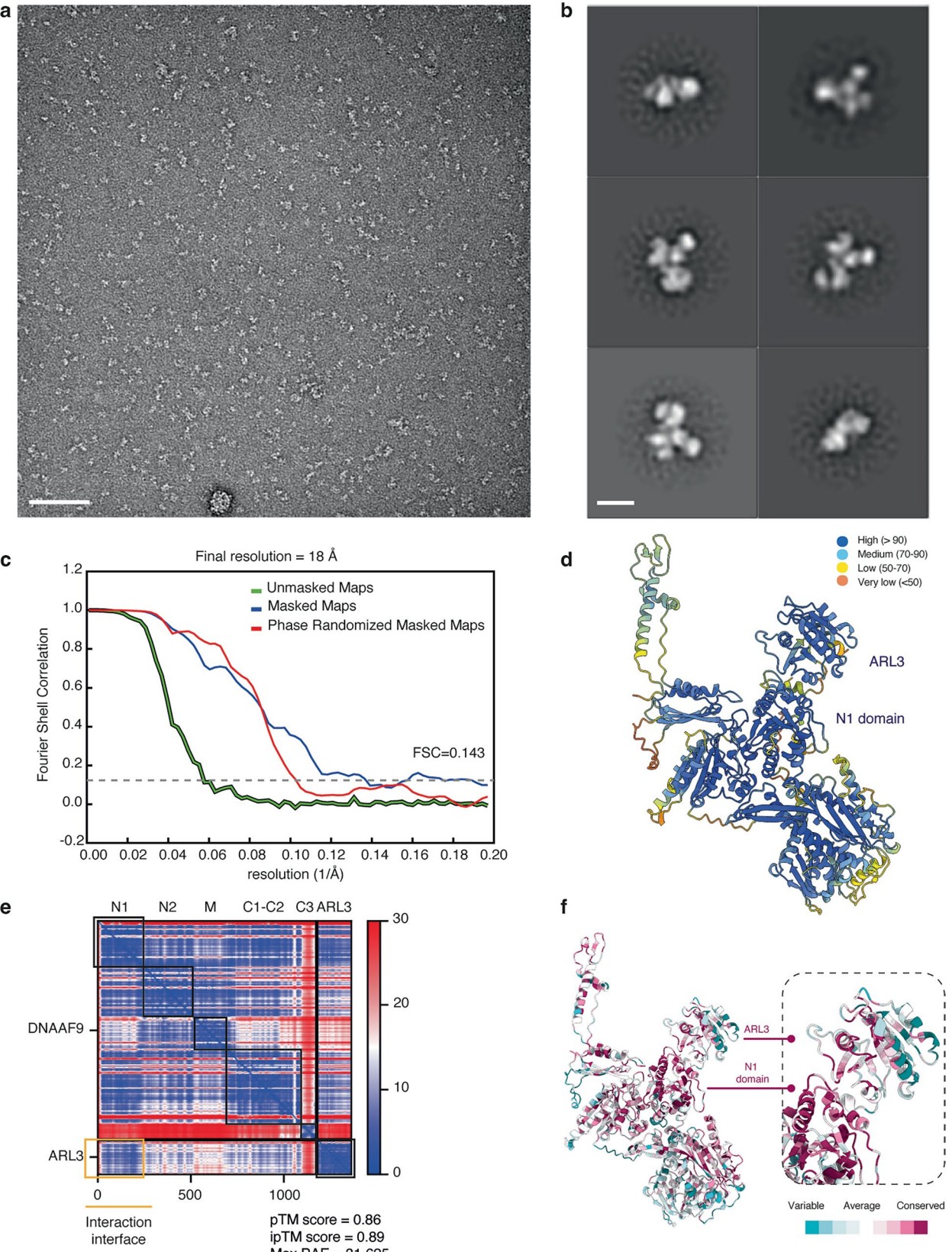

**Extended Data Fig. 8 | Negative stain EM, AlphaFold2-Multimer and CONSURF analysis of the DNAAF9-ARL3 complex. a**. A representative negative stain electron micrograph showing single particles from a DNAAF9-Arl3$^{Q70L}$ reconstitution experiment is shown (n = 3 replicates). Scale bar, 100 nm. **b**. Representative class averages of DNAAF9 bound by Arl3$^{Q70L}$ from 2D classification of the reconstituted complex. Scale bar, 2.5 nm **c**. Gold standard Fourier shell correlation (FSC) curves for DNAAF9-Arl3 complex map (Fig. 4b)

as determined by RELION-4.0 (FSC = 0.143). **d**. pLDDT scores are plotted onto the predicted DNAAF9-ARL3 structure to highlight local confidence in the modeling (dark blue = pLDDT 90-100; high confidence; orange = pLDDT 0-50; low confidence). **e**. PAE plot from AF2-Multimer prediction of the DNAAF9-ARL3 complex. The interaction interface is highlighted, and model statistics are shown. **f**. CONSURF scores mapped onto the predicted model of the DNAAF9-ARL3 complex.

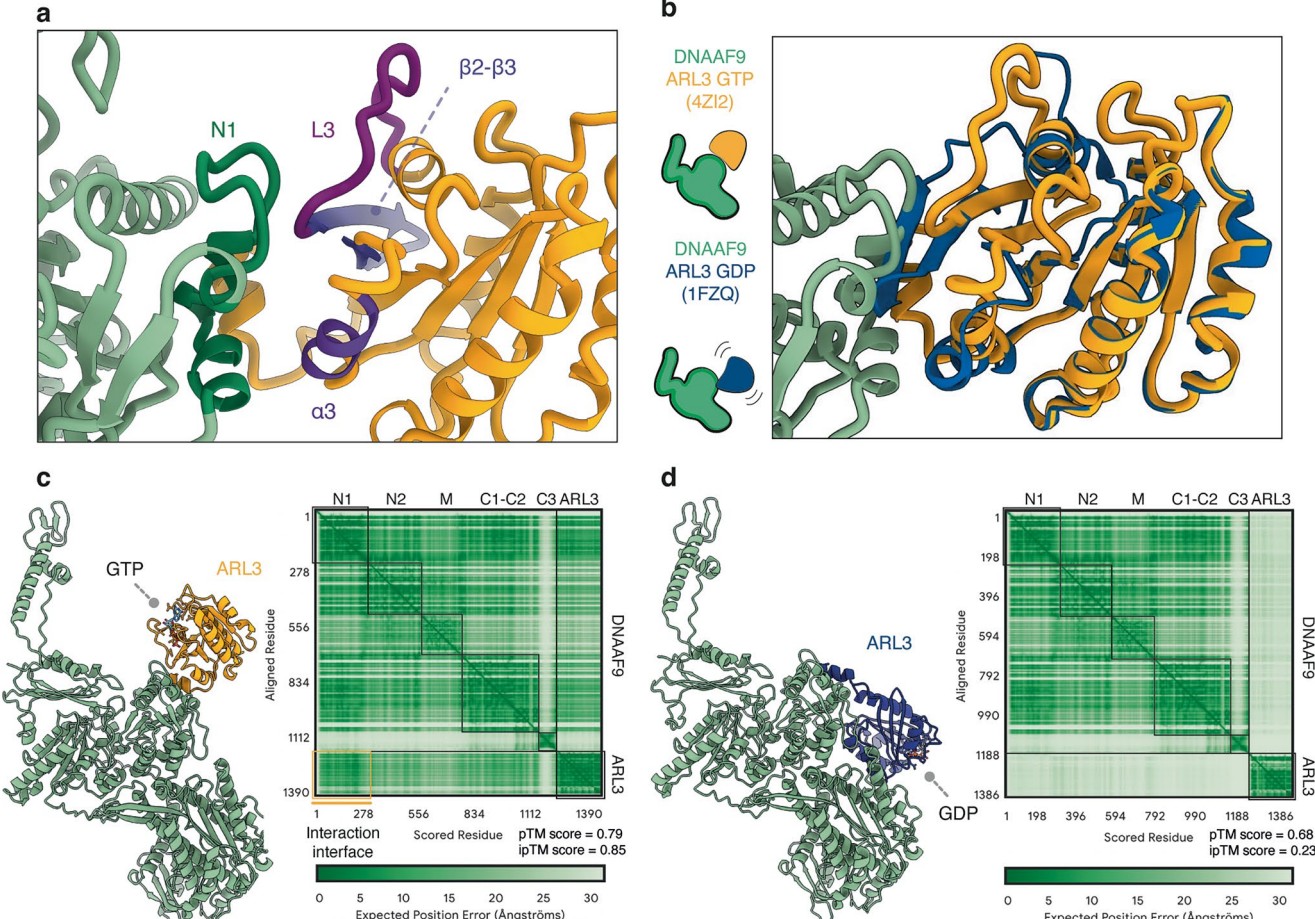

**Extended Data Fig. 9 | DNAAF9-ARL3^GDP complex formation is hindered by steric clashes. a**. Secondary structure features in ARL3's GTP loaded state that interact with DNAAF9's N1 domain are shown. **b**. Structure of ARL3 in its GDP loaded state (ARL3-GDP in blue from pdb: 1FZQ) superimposed on the AF2-Multimer prediction of the DNAAF9-ARL3 complex (ARL3-GTP in mustard; pdb: 4ZI2). ARL3's L3 loop and β2 strand adopts a beta-hairpin in the GDP state

creating steric clashes between ARL3-GDP and DNAAF9 that are incompatible for stable complex formation. **c, d**. AF3-Multimer models of DNAAF9-ARL3 in the presence of GTP and GDP ligands. PAE plots are shown to the right of each prediction. The interaction interface is highlighted for (**c**), and model statistics are shown.

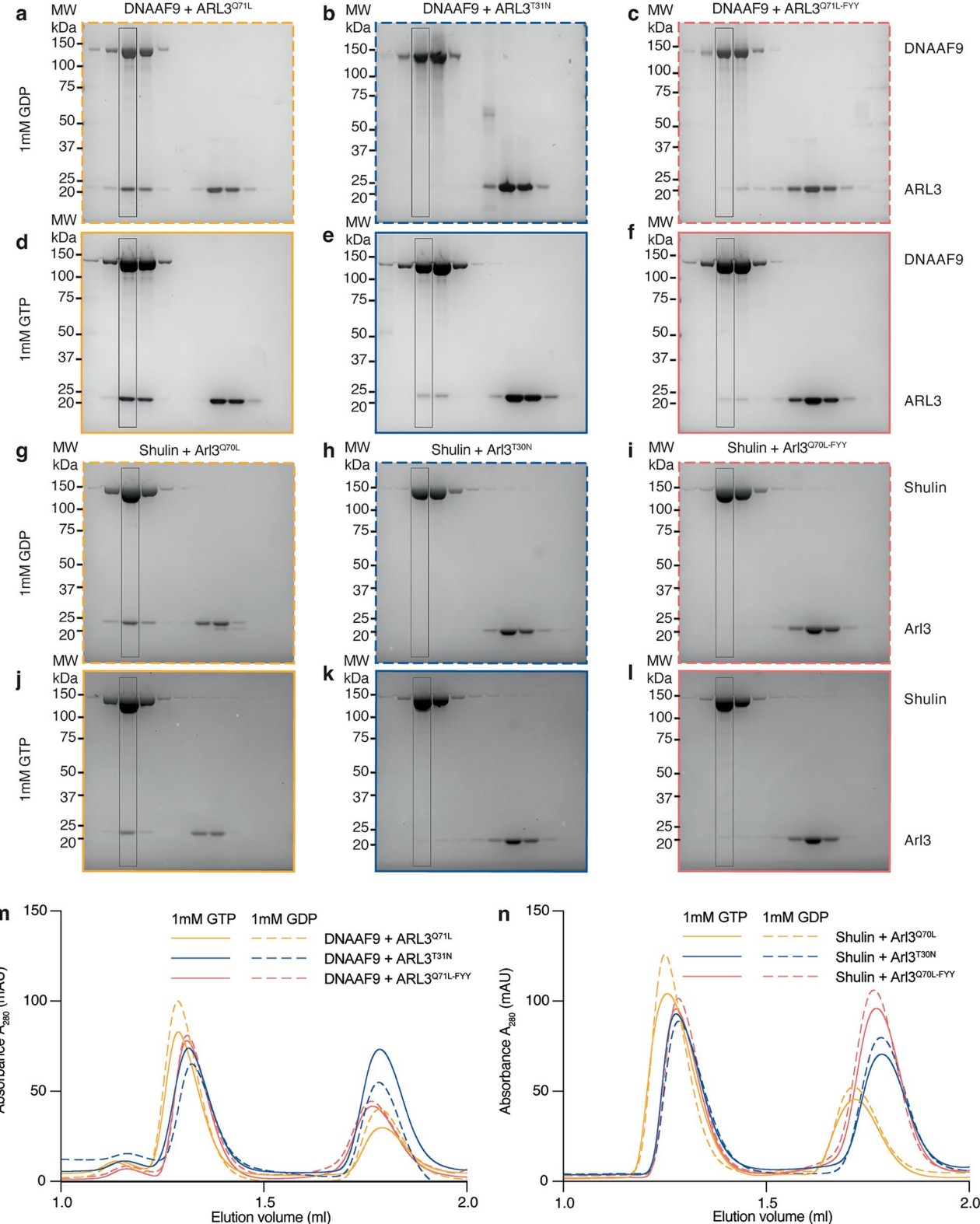

**Extended Data Fig. 10 | DNAAF9 (Shulin) forms a stable complex with GTP-locked ARL3 (Arl3). a-f, m**. SDS-PAGE gels and corresponding analytical size exclusion chromatography profiles of reconstitutions between recombinant DNAAF9 and three ARL3 variants [GTP-locked Q71L, GDP-mimic T31N and interface FYY mutant on a Q71L background] in 1 mM GTP or 1 mM GDP, resolved over a Superdex 200 3.2/300 column are shown. **g-l, n**. Reconstitutions of Shulin and three Arl3 variants [GTP-locked Q70L, GDP-mimic T30N and interface FYY mutant on a Q70L background] in 1 mM GTP or 1 mM GDP, resolved over a Superdex 200 3.2/300 column are shown. In each of the reconstitution gel images, a box around one of the peak fractions highlights the presence or absence of a stable complex forming. Reconstitutions were performed twice with similar results. Molecular weight markers (MW) are shown in kDa.

# Reporting Summary

## Statistics

For all statistical analyses, confirm that the following items are present in the figure legend, table legend, main text, or Methods section.

| n/a | Confirmed | |
|---|---|---|
| ☐ | ☒ | The exact sample size (*n*) for each experimental group/condition, given as a discrete number and unit of measurement |
| ☐ | ☒ | A statement on whether measurements were taken from distinct samples or whether the same sample was measured repeatedly |
| ☐ | ☒ | The statistical test(s) used AND whether they are one- or two-sided<br>*Only common tests should be described solely by name; describe more complex techniques in the Methods section.* |
| ☒ | ☐ | A description of all covariates tested |
| ☐ | ☒ | A description of any assumptions or corrections, such as tests of normality and adjustment for multiple comparisons |
| ☐ | ☒ | A full description of the statistical parameters including central tendency (e.g. means) or other basic estimates (e.g. regression coefficient) AND variation (e.g. standard deviation) or associated estimates of uncertainty (e.g. confidence intervals) |
| ☐ | ☒ | For null hypothesis testing, the test statistic (e.g. *F*, *t*, *r*) with confidence intervals, effect sizes, degrees of freedom and *P* value noted<br>*Give P values as exact values whenever suitable.* |
| ☒ | ☐ | For Bayesian analysis, information on the choice of priors and Markov chain Monte Carlo settings |
| ☒ | ☐ | For hierarchical and complex designs, identification of the appropriate level for tests and full reporting of outcomes |
| ☒ | ☐ | Estimates of effect sizes (e.g. Cohen's *d*, Pearson's *r*), indicating how they were calculated |

*Our web collection on statistics for biologists contains articles on many of the points above.*

## Software and code

Policy information about availability of computer code

| Data collection | Leica SP8 Confocal Laser Scanning Microscope, Olympus FV3000 upright confocal microscope, Thermo iBright FL1000 Imaging System, FEI EPU on an FEI 200kV Tecnai 20 TEM, JEOL 2100 Plus 200kV TEM. |
|---|---|
| Data analysis | Main software or servers used for data analyses are publicly available, detailed in the manuscript methods section and included below: Fiji v2.1.0/1.53c, GraphPad Prism v10, RELION v3.2 or v4.0, UCSF ChimeraX v1.7-v1.9, Google Colab Alphafold2 Notebook (https://colab.research.google.com/github/sokrypton/ColabFold/blob/main/AlphaFold2.ipynb) and Alphafold 3 (AlphaFold web server: https://alphafoldserver.com/), PSI-BLAST, Clustal Omega (integrated in Uniprot: https://www.uniprot.org/), ESPript 3 (https://espript.ibcp.fr/ESPript/ESPript/), CONSURF server (https://consurf.tau.ac.il/consurf_index.php). |

For manuscripts utilizing custom algorithms or software that are central to the research but not yet described in published literature, software must be made available to editors and reviewers. We strongly encourage code deposition in a community repository (e.g. GitHub). See the Nature Portfolio guidelines for submitting code & software for further information.

## Data

Policy information about availability of data

All manuscripts must include a data availability statement. This statement should provide the following information, where applicable:

- Accession codes, unique identifiers, or web links for publicly available datasets
- A description of any restrictions on data availability
- For clinical datasets or third party data, please ensure that the statement adheres to our policy

Mass spectrometry data have been deposited in ProteomeXchange and made available via the PRIDE48 partner repository with the primary accession code PXD052722. Source data have been provided in Source Data and also available at https://doi.org/10.6084/m9.figshare.27195438. All other data supporting the findings of this study are available from the corresponding author upon reasonable request.

## Research involving human participants, their data, or biological material

Policy information about studies with human participants or human data. See also policy information about sex, gender (identity/presentation), and sexual orientation and race, ethnicity and racism.

| Reporting on sex and gender | N/A |
|---|---|
| Reporting on race, ethnicity, or other socially relevant groupings | N/A |
| Population characteristics | N/A |
| Recruitment | N/A |
| Ethics oversight | N/A |

Note that full information on the approval of the study protocol must also be provided in the manuscript.

# Field-specific reporting

Please select the one below that is the best fit for your research. If you are not sure, read the appropriate sections before making your selection.

☒ Life sciences        ☐ Behavioural & social sciences        ☐ Ecological, evolutionary & environmental sciences

For a reference copy of the document with all sections, see nature.com/documents/nr-reporting-summary-flat.pdf

# Life sciences study design

All studies must disclose on these points even when the disclosure is negative.

| Sample size | Human airway epithelial cells (HAECs) were obtained as transwell cultures containing separate pooled batches of cells from Epithelix Sarl at two differentiation time-points. Cell clumps were gently scraped from transwell inserts and suspended down to single cells which were then applied to glass slides and allowed to air-dry. Each slide contained several hundred individual cells which were processed for immunostainings. Many cells were observed under a confocal microscope. Of these 10-20 representative cell images/batch were randomly acquired for quantitative analyses. |
|---|---|
| | All zebrafish localization experiments were performed in at least 2 biological replicates and the numbers of control and injected embryos imaged were n>4. |
| | Sample sizes were not pre-determined for the structural analyses performed in this study. Negative stain EM data was collected until a sufficiently high resolution to answer our biological question was achieved. |
| Data exclusions | No data were excluded from the cell immunostaining analyses. EM micrographs that contained empty areas and thick stain in the field of view were excluded. During 2D and 3D classification, particles that did not yield detailed images to reveal structural densities were also excluded. |
| Replication | Immunostainings performed on three replicate batches showed similar sub-cellular distributions for endogenous DNAAF9 and DNAI2 in the immature and more mature human airway cells. |
| | IP-MS experiments were performed in triplicate followed by 12-plex TMT mass spectrometry runs. |
| | ODA-Shulin biochemical reconstitutions and displacement experiments using Arl3 Q70L or Arl3 Q70L-FYY were performed in triplicate and these replicate data were used for gel densitometry based quantification analyses. |
| | Imaging of n>4 control and injected embryos from 2 biological replicates showed similar sub-cellular distribution for the C-terminal tagged versions of Dnaaf9 protein. |

No attempt was made to replicate the negative stain EM data which involves averaging of several thousands of individual protein molecules.

| | |
|---|---|
| Randomization | Cell images from randomly selected areas were acquired to allow for unbiased random sampling. ~10-20 cell images from each batch were acquired to provide data from three biologically different batches. Randomization was not relevant to the structural methods of this study. |
| Blinding | Three study authors (BB, MR and GRM) were involved in acquiring cell images independently. It was difficult to perform blinding as the cell slides were labeled and additional blinding was not attempted. |

# Reporting for specific materials, systems and methods

We require information from authors about some types of materials, experimental systems and methods used in many studies. Here, indicate whether each material, system or method listed is relevant to your study. If you are not sure if a list item applies to your research, read the appropriate section before selecting a response.

## Materials & experimental systems

| n/a | Involved in the study |
|---|---|
| ☐ | ☒ Antibodies |
| ☐ | ☒ Eukaryotic cell lines |
| ☒ | ☐ Palaeontology and archaeology |
| ☐ | ☒ Animals and other organisms |
| ☒ | ☐ Clinical data |
| ☒ | ☐ Dual use research of concern |
| ☒ | ☐ Plants |

## Methods

| n/a | Involved in the study |
|---|---|
| ☒ | ☐ ChIP-seq |
| ☒ | ☐ Flow cytometry |
| ☒ | ☐ MRI-based neuroimaging |

## Antibodies

| | |
|---|---|
| Antibodies used | Primary antibodies commercial: DNAAF9 (Proteintech, 23184-1-AP) DNAI2 (Abnova, H00064446-M01, clone IC8), Acetylated alpha tubulin  (Cell Signaling, 5335), IFT74 (Proteintech, 27334-1-AP), IFT81 (Proteintech, 11744-1-AP), anti-Strep tag (GT661, Thermo Scientific, MA5-17283), Myc-tag (Santa Cruz, sc-40), GFP (Abcam, ab13970).<br><br>Primary antibodies custom generated: Tetrahymena ODA holocomplex (Eurogentec).<br><br>Secondary antibodies commercial: Alexa Fluor 568 anti-rabbit antibody (Invitrogen, A-11011), Alexa Fluor 488 anti-mouse antibody (Invitrogen, A-10680), Alexa 555 anti-mouse antibody (Invitrogen, A-28180), Alexa 488 anti-rabbit antibody (Invitrogen, A-11034). GeneTex, EasyBlot anti-rabbit IgG and anti-mouse IgG HRP conjugated monoclonal antibodies; GTX221666-01-S and GTX221667-01-S respectively). |
| Validation | Commercially available antibodies have validation and relevant citation data on the manufacturers website. DNAAF9 antibody was additionally validated in house by immunoblotting for purified recombinant DNAAF9 protein and by immunoprecipitation and mass spectrometric detection of target protein i.e. DNAAF9 as the top hit. DNAI2 antibody was previously tested in Diggle et al., DOI: 10.1371/journal.pgen.1004577.<br><br>Custom antibody raised to the Tetrahymena ODA holocomplex was previously tested and used in Mali et al., 2021; DOI: 10.1126/science.abe0526. |

## Eukaryotic cell lines

Policy information about cell lines and Sex and Gender in Research

| | |
|---|---|
| Cell line source(s) | Sf9 cells were obtained from the Eukaryotic Expression Facility (EEF) operated by the Berger lab at the University of Bristol's School of Biochemistry.<br><br>Human airway epithelial cells (HAECs) were obtained as transwell cultures containing separate pooled batches of cells from multiple donors from Epithelix Sarl (Switzerland). |
| Authentication | None of the cell lines were authenticated by the study authors. All cell lines or primary cells were obtained from reliable sources with in-house quality control procedures in place. |
| Mycoplasma contamination | Cell lines tested negative for mycoplasma contamination. |
| Commonly misidentified lines (See ICLAC register) | N/A |

# Animals and other research organisms

Policy information about studies involving animals; ARRIVE guidelines recommended for reporting animal research, and Sex and Gender in Research

| | |
|---|---|
| Laboratory animals | Mice, Zebrafish |
| Wild animals | Study did not involve wild animals. |
| Reporting on sex | Both male and female animals were equally analysed. 8 male and 8 female adult mutant animals were phenotyped. Brain histopathology was performed on 2 male and 2 female homozygote adult animals. Motile cilia are fundamental to the physiology of multiciliated cells and it is unlikely that the findings would apply specifically to only one sex. |
| Field-collected samples | Study did not involve samples collected from the field. |
| Ethics oversight | Mice - All procedures on animals at The Centre for Phenogenomics (TCP) were reviewed and approved by TCP's Animal Care Committee. TCP is certified by the Canadian Council on Animal Care and registered under the Animals for Research Act of Ontario. TCP is part of the International Mouse Phenotyping Consortium.<br><br>Zebrafish - All experiments with zebrafish embryos were approved by the Singapore National Advisory on Laboratory Animal Research. |

Note that full information on the approval of the study protocol must also be provided in the manuscript.

# Plants

| | |
|---|---|
| Seed stocks | N/A |
| Novel plant genotypes | N/A |
| Authentication | N/A |

