## [Peer Review File · Nature Structural & Molecular Biology]

Molecular basis for the activation of outer dynein arms in cilia

Corresponding Author: Dr Girish Mali

Version 0:

Decision Letter:

30th Oct 2024

Dear Dr. Mali,

Thank you for submitting your manuscript "Molecular basis for the activation of outer dynein arms in cilia". We have now carefully evaluated the work and discussed it among the editorial team. Unfortunately, we have decided not to consider the manuscript further for publication in Nature Structural & Molecular Biology.

We can only consider a small proportion of the manuscripts submitted to our journal and are often forced to make difficult decisions. Manuscripts are evaluated editorially for their potential interest to a broad audience, the level of novel insight obtained and whether the findings represent a significant advance relative to the published literature, among other considerations.

In this case, we are interested in this area of research and appreciate the potential role for a Shulin-Arl3 complex in regulating Shulin and its rebinding to ODAs. We recognize that the findings will be of value to others working in this area. However, after discussion among the editorial staff, I am afraid we are not persuaded that the level of mechanistic insight and degree of conceptual advance obtained warrants publication in Nature Structural & Molecular Biology, after taking into account relevant published work [including, but not limited to, studies suggesting that DNAAF9 may bind to ARL3; other work showing that Arl3 binds to another potential ODA transport adaptor, Oda16, with Oda16 proposed to act as an effector in ciliary transport]

Although we cannot offer to publish your manuscript, I suggest that you consider Nature Communications as a suitable venue for this work. To transfer your manuscript, please use our manuscript transfer portal. You will not have to re-supply manuscript metadata and files, unless you wish to make modifications. For more information, please see our [manuscript transfer FAQ](http://www.nature.com/authors/author_resources/transfer_manuscripts.html?WT.mc_id=EMI_NPG_1511_AUTHORTRANSF&WT.ec_id=AUTHOR) page.

I am sorry we could not be more positive on this occasion. We thank you for the opportunity to consider this work and wish you success in seeking publication elsewhere.

Sincerely,

Melina Casadio, PhD
Consulting Editor, Nature Structural & Molecular Biology
Senior Editor, Nature Cell Biology
ORCID ID: <https://orcid.org/0000-0003-2389-2243>

** For Springer Nature Limited general information and news for authors, see <http://npg.nature.com/authors>.

Version 1:

Decision Letter:

4th Dec 2024

Dear Dr. Mali,

Thank you for your letter concerning your manuscript "Molecular basis for the activation of outer dynein arms in cilia". We really appreciate your patience with the process and are sorry we cannot prioritize appeals over new submissions. We have now had a chance to discuss the points you raised in detail, and we have decided to send your paper out to review. Prior to peer review, please see the points below to prepare the files necessary for peer review:

1- We want to ensure that the methods and statistics reporting in our papers are of the highest quality. To that end, we ask authors to fill out a Reporting Summary that collects information on experimental design and reagents, as well as an editorial Policy Checklist, which confirms compliance with our editorial policies, including the declaration of Competing Interests.

These documents can be found by following the links below:

Reporting Summary:

Editorial Policy Checklist:

<https://www.nature.com/documents/nr-editorial-policy-checklist.pdf>

Please complete the relevant forms and include them with your files for peer review. Please note that these forms are dynamic 'smart pdfs' and must, therefore, be downloaded and completed in Adobe Reader. We will then flatten them for ease of use by the reviewers. If you would like to reference the guidance text as you complete the template, please access these flattened versions at <http://www.nature.com/authors/policies/availability.html>.

Note that you are not required to revise your paper to include the information provided in the reporting summary. However, all points on the policy checklist must be addressed; please include a new version of the manuscript with your completed checklist if needed.

Once we receive these documents and review them to ensure that all requested information is provided, we will proceed to send your paper for review. If you have questions or anticipate delays, please let me know as soon as possible.

2- Proteomics datasets should be deposited and made available to the reviewers at a minimum. Please clarify where the mass spectrometry datasets can be accessed with your resubmission.

3- When preparing your revised manuscript including the appeal data, please note that, if any personal communications are included, you would need to forward an email permission to us stating that the persons who made these communications to you approve their inclusions in this particular manuscript. Please let me know if you have any questions.

You can use the link below to be taken directly to the site and submit your manuscript:

Link Redacted

Sincerely,

Melina

Melina Casadio, PhD
Consulting Editor, Nature Structural & Molecular Biology
Senior Editor, Nature Cell Biology
ORCID ID: <https://orcid.org/0000-0003-2389-2243>

Version 2:

Decision Letter:

24th Jan 2025

Dear Dr Mali,

Thank you again for submitting your manuscript "Molecular basis for the activation of outer dynein arms in cilia". I apologize for the delay in responding, which resulted from the difficulty in obtaining suitable referee reports. We now have comments (below) from the 3 reviewers who evaluated your paper. In light of those reports, we remain interested in your study and would like to see your response to the comments of the referees in the form of a revised manuscript.

To guide the scope of the revisions, we list below a prioritized set of referee points that should be addressed in the revision,

which we hope will be helpful to you. You will see that the reviewers found the model interesting but asked for additional evidence to support it, as well as clarifications:

-- In revision, please strengthen the mechanism of Shulin-ODA complex regulation by Arl3 (Rev#2 point #2 – please also see Rev#1 point #1 to strengthen the functional implications for DNAAF9) and the studies characterizing the binding between DNAAF9 and Arl3 (Rev#2 point #1)

-- Please address all minor points and suggestions to strengthen the current dataset including Rev#1's point #2; please address Rev#1's questions in point #3 about the various models used.

We are committed to providing a fair and constructive peer-review process. Do not hesitate to contact us if there are specific requests from the reviewers that you believe are technically impossible or unlikely to yield a meaningful outcome. Please be sure to address/respond to all concerns of the referees in full in a point-by-point response and highlight all changes in the revised manuscript text file. If you have comments that are intended for editors only, please include those in a separate cover letter.

Our standard revision period is 3 to 6 months. If you cannot send it within this time, please let us know. We will be happy to consider your revision as long as nothing similar has been accepted for publication at NSMB or published elsewhere. Should your manuscript be substantially delayed without notifying us in advance and your article is eventually published, the received date would be that of the revised, not the original, version.

Reporting Summary:

When submitting the revised version of your manuscript, please pay close attention to our [href="https://www.nature.com/nature-portfolio/editorial-policies/image-integrity">Digital Image Integrity Guidelines. and to the following points below:](https://www.nature.com/nature-portfolio/editorial-policies/image-integrity)

EXTENDED DATA FIGURES

Please note that all key data shown in the main figures as cropped gels or blots should be presented in uncropped form, with molecular weight markers. These data can be aggregated into a single supplementary figure. While these data can be displayed in a relatively informal style, they must refer back to the relevant figures. These data should be submitted with the last revision, prior to acceptance, but you may want to start putting it together at this point.

We require deposition of coordinates (and, in the case of crystal structures, structure factors) into the Protein Data Bank with

the designation of immediate release upon publication (HPUB). Electron microscopy-derived density maps and coordinate data must be deposited in EMDDB and released upon publication. Deposition and immediate release of NMR chemical shift assignments are highly encouraged. Deposition of deep sequencing and microarray data is mandatory, and the datasets must be released prior to or upon publication. To avoid delays in publication, dataset accession numbers must be supplied with the final accepted manuscript and appropriate release dates must be indicated at the galley proof stage. Please find the complete NRG policies on data availability at <http://www.nature.com/authors/policies/availability.html>.

Link Redacted

Sincerely,

Melina Casadio, PhD
Locum Chief Editor, Nature Structural & Molecular Biology
ORCID ID: <https://orcid.org/0000-0003-2389-2243>

Referee expertise:

Referee #1: dynein

Referee #2: dynein, structural biology

Referee #3: cilia

Reviewers' Comments:

Reviewer #1 (Remarks to the Author):

This is an excellent manuscript with very clearly stated hypotheses and satisfying answers to the questions posed. In brief, the authors follow up on a 2021 manuscript in which they initially described Shulin's biochemical and biological activity along with a high resolution cryo-EM structure. They speculate that DAAAF9 is a likely homolog to the T. thermophila Shulin protein, which their previous work demonstrated was an important effector of OAD's autoinhibited state. They find that DAAAF9 localizes to the developing cilia of human airway epithelial cells (but not to mature cilia), co-IPs with OAD components (eg. DNAI2), and its addition to purified OAD causes a change in conformation to a 'closed'/presumably autoinhibited state. The authors then focus on testing the hypothesis that the Arl3 GTPase releases ODA from DAAAF9 in a GTP-dependent manner. They find Arl3 from human and T. thermophila both bind to DAAAF9/Shulin. Using negative stain EM and AF2, they map the approximate contact point between Arl3 and DAAAF9 that they confirm via mutagenesis and binding assays. Finally, the authors show that a GTP-bound Arl3 indeed has the capacity to dissociate a T. thermophila Shulin-ODA complex. I strongly support publication of the manuscript providing they address a few concerns I have, which may help to improve the manuscript prior to publication.

Major concerns:

- 1) With regard to "Taken together, these findings indicate that human DAAAF9 interacts with mammalian ODAs and maintains them in a closed conformation sharing its role as an inhibitor with its Tetrahymena ortholog Shulin", I wonder why the authors didn't test DAAAF9 in a gliding assay as they did previously (Mali et al., Science 2021)? Although the negative stain EM evidence is fairly strong, I'm not sure their findings "indicate" a role for DAAAF9 in inhibiting ODAs. Such an experiment would provide much stronger evidence that DAAAF9 indeed performs an analogous role to Shulin in ODA function. It would also provide support for the "inhibitory role" proposed in this sentence from the Discussion: "...we favour a model where the critical inhibitory role played by Shulin (DAAAF9) is central to stabilising IFT-ODA interactions for efficient transport into cilia." Without such evidence, the language would need to be changed to reflect this.
- 2) Although I'm okay with the model in Figure 2i being presented as is, the data to support it is a bit thin (quantitative MS data). In particular, the model for dynein-1 transport of DAAAF9, although intriguing, may need more support. Do the authors see any cargo adaptor proteins in the MS data? Although not required, this would make this argument more convincing.

Also, have the authors tried looking for dynein-1 complex components in their EC vs MC pull-downs?

3) I was a little confused by the back-and-forth between model systems. I think it's great that the authors included all these data, but my head started to spin by the last figure in particular. It would help if there was more explanation as to why they performed some experiments with human components, and others with *T. thermophila* proteins. For example, why didn't the authors test whether human Arl3 induces dissociation of the DAAF9-ODA complex?

Minor points:

- 1) Although I was able to guess which was which, the AF2 models for Shulin and DAAF9 in Fig. S1 are not labeled. Please add labels.
- 2) It seems that the sequence for *T. thermophila* Shulin is missing from Figure S2. If not, please label accordingly. If so, I think this would be a helpful point of comparison.
- 3) The data presented in the Discussion should either be moved to the Results section (and described in detail) or removed from the paper entirely (and saved for a future paper). This includes Fig. S9 and 10.
- 4) The paragraphs in the Discussion describing Oda16 and potential transport adaptors (line 366-390) seemed off-topic, and was a distraction to me as a reader. I had trouble finding out how this fit into the paper. I feel the same is true for their argument that DAAF9 is a transport adaptor for IFT. I would suggest removing this part of the discussion and accompanying data in Figure S10 (save it for another paper) and simply focus on discussing the data within the results section.
- 5) "Given these emerging links between ARL3 and the biogenesis of motile cilia, we recommend that primary ciliopathy patients carrying ARL3 variants should be additionally assessed for clinical features indicative of defective motile cilia." Maybe this is born from ignorance, but wouldn't primary ciliopathy patients already be assessed for potential clinical features of defective motile cilia?

Reviewer #2 (Remarks to the Author):

Issa et al. set out to determine the functional consequence of the interaction between DAAF9 and the small G protein Arl3. The authors find that DAAF9 (and its tetrahymena homolog Shulin) preferentially bind to GTP-bound Arl3. Arl3-DAAF9 interaction displaces DAAF9 from outer dynein arm complex, thus resulting in dynein activation in cilia. The motivation for this work is important, as ciliary beating defects causes ciliopathies and the findings were described clearly. Pending two important points below, I recommend this work for publication.

Major points:

1. The authors use mutagenesis to try to validate the predicted interface between DAAF9 and Arl3 (they make three mutations: F51A, Y77A, and Y81A). However, these mutations should be made in the Arl3-Q70L GTP-locked background, not the WT background. The data in Figure 3A shows that WT Arl3 that can GTP cycle doesn't really bind DAAF9 (maybe there is a little shoulder that shows complex co-elution, but the interaction is not robust enough to see Arl3 on the SDS-PAGE gel in b). The authors should test if the Arl3- F51A, Y77A, Y81A mutant in the Q70L background has reduced affinity for DAAF9.
2. The data in Figure 5 is very important to the authors model but is the least convincing in the paper. The following points are important to fortify this data:
 - a. The data in Figure 5d must be quantified, rather than just shown qualitatively. For example, the authors could measure the intensity of the bands of ODA heavy chain: Shulin in peak complex fractions with and without inclusion of Arl3. In fact, it is actually very challenging to tell if inclusion of Arl3 really does reduce Shulin-ODA interaction because there is significantly more total protein loaded on the gel for the experiment without Arl3.
 - b. The GTPase dead Arl3 mutant or the Arl3 mutant variants that don't bind (but in the GTP-locked background) would be a great negative control for this study.
 - c. Finally, the authors should show the SEC traces for the experiments with and without Arl3, as well as show gels that have the exact same fractions (and where they correspond to the SEC elution profile) for each experiment. It is hard to assess the quality of this data without seeing the traces and all the same corresponding fractions.

Minor points:

1. Zoom in on figure 1d would be helpful to show the DAAF9 staining on the cilia.
2. Can the authors report the % open and % closed ODA conformations with and without DAAF9 (data in figure S5) to quantify the effect of DAAF9 on ODA structure?

Reviewer #3 (Remarks to the Author):

Review of NSMB-A49966B

The manuscript by Issa et al. from the Mali lab addresses a fundamental question in ciliary biology. More specifically, how

are the multimeric axonemal dyneins assembled and maintained in the cytoplasm in an inactive form and then activated once they enter cilia and flagella and attach to the outer doublet microtubules. In previous work the authors identified a dynein chaperone known as Shulin that maintains outer dynein arms (ODAs) in an inactive conformation in the cytoplasm and co-localizes with ODAs in regenerating *Tetrahymena* cilia. In the present study they ask (1) whether this inhibitory mechanism is conserved in Shulin orthologs such as the human DNaAF9, (2), whether it interacts with transport factors to target ODAs to cilia, and (3) how is Shulin/DNaAF9 released from ODAs inside cilia to allow ODA activation?

Most significant findings:

Other studies had shown that DNaAF9 could be pulled down with the small GTPase ARL3, which suggested that DNaAF9 might be involved in the ciliary transport of ODAs. The authors used AlphaFold 2 (AF2) analyses to compare the structures and sequences of Shulin and DNaAF9 (Figures S1, S2). These analyses identified several conserved domains with significant structural similarity that could be involved in binding to ODAs during transport. The authors test this model using human airway epithelial cells cultured at an air-liquid interface to induce differentiation of basal cells into multiciliated cells (Figure 1). Consistent with their hypothesis that Shulin and DNaAF9 share a similar function, they observed that DNaAF9 enters growing motile cilia in differentiating airway cells (Fig 1d, e). Similar results were observed using zebrafish embryos (Figure S3).

To gain a better understanding into the mechanism by which DNaAF9/Shulin might facilitate the transport of ODAs, they used immunoprecipitation, TMT labeling, and mass spectrometry to identify interacting proteins (Figure 2, Supplemental Datafile 1). This identified more than 87 proteins found at different stages of differentiation, including ODA subunits, IFT-B subunits, ciliary tip proteins, and ARL3. The proteomics work appears to be rigorous, and the major take-homes are nicely summarized in their Figure 2, but I would have liked to have seen more detailed explanation of their results in the Supplemental Datafile 1. If I understand their methods correctly, the IPs were performed using total cell lysates and not purified cilia. Thus, the model proposed in Figure 2i is an inference based on the results from total cell lysates. However, they test several key points of their model using a clever series of *in vitro* binding experiments.

To test the hypothesis that ARL3 might regulate that activity of DNaAF9 and Shulin, they expressed recombinant DNaAF9 and *Tetrahymena* ARL3 in its GTP and GDP mimicking states using Q70L and T30N variants respectively. They found that all Arl3 variants co-eluted with DNaAF9 in high GTP (Fig. S6) but only Arl3Q70L binds DNaAF9 in the presence of GDP (Figure 3). Using this Arl3 variant as bait and lysates of *Tetrahymena* cells undergoing ciliary regeneration, they co-immunoprecipitated several known effectors of Arl3 and Shulin (Figure 3), suggesting that Arl3 is likely to regulate transport of ODAs through its interaction with Shulin.

The authors extended this work by analyzing the interaction between recombinant DNaAF9 (or Shulin) in complex with Arl3Q70L *in vitro* by negative stain electron microscopy (Figure 4, Fig. S7). Single particle averaging identified a subset of images (~50%) containing a DNaAF9/Shulin-Arl3 complex, with the ARL3 subunit bound to the N1 domain of DNaAF9/Shulin. Analytical chromatography confirmed that this interaction was most stable using the Arl3Q70L variant. They extended these observations using AF2 to model the interaction interface and then disrupted the interaction using specific ARL3 mutations (Figure 4e). Further modeling using AF3 suggests that ARL3 interacts with DNaAF9 in the presence of GTP but not GDP (Fig. S8). Additional *in vitro* studies demonstrate that ARL3 can disrupt the interaction between purified Shulin and ODAs in the presence of GTP (Figure 5).

Overall Evaluation and Significance

In summary, the authors combine sophisticated modeling, well executed proteomics, and elegant *in vitro* studies to test a new model of how the ARL3 GTPase might regulate the interaction between a dynein chaperone and the ODAs to maintain the ODAs in an inactive state in the cytoplasm and release and activate the ODAs upon entry into the cilia. The experimental logic is clearly described, the experiments are well executed, and the suggested mechanism of regulation for dynein activation *in vivo* is a novel, exciting hypothesis. Future *in vivo* studies will be needed to determine the precise timing and location of ODA activation, but the work presented here is an exciting step forward. I think the work will be of broad interest to people working in the field of motor activation and targeting, and I highly recommend publication.

Version 3:

Decision Letter:

23rd May 2025

Dear Girish,

Thank you again for alerting us to the issue with data in S3df in your manuscript "Molecular basis for the activation of outer dynein arms in cilia".

I have now discussed this matter with our Editorial Director.

We thank you for bringing to our attention the use of an incorrect myc construct in experiments underlying the data in S3df. Following our internal assessment and given the need to provide new data and remove present data, we regret that we are unable to proceed with the acceptance in principle of the current version of the manuscript.

We are open to reconsidering the manuscript on the condition that revised data in full compliance with our policies are provided. Any decision to proceed with such a revised manuscript will depend on further assessment of these data editorially and might involve reinitiating the peer-review process to determine whether the level of support for the interpretation of the data and the conclusions drawn remains unaltered.

Please also pay close attention to our image integrity and imaging standards policies <https://www.nature.com/nature-portfolio/editorial-policies/image-integrity>

Please provide a document (e.g., cover letter) clarifying the issues detected for particular figure panels (S3df and any other) in the current set of figures (version NSMB-A49966C) and how these issues were addressed in the revised manuscript.

Please note that you will also need to re-upload with your final resubmission all source data files (source numerical data, source blots) and the reporting summary.

Reporting Summary:

When submitting the revised version of your manuscript, please pay close attention to our [href="https://www.nature.com/nature-portfolio/editorial-policies/image-integrity">Digital Image Integrity Guidelines.](https://www.nature.com/nature-portfolio/editorial-policies/image-integrity) and to the following points below:

EXTENDED DATA FIGURES

Please note that all key data shown in the main figures as cropped gels or blots should be presented in uncropped form, with molecular weight markers. These data can be aggregated into a single supplementary figure item. While these data can be displayed in a relatively informal style, they must refer back to the relevant figures. These data should be submitted with the final revision, as source data, prior to acceptance, but you may want to start putting it together at this point.

Data availability: this journal strongly supports public availability of data. All data used in accepted papers should be available via a public data repository, or alternatively, as Supplementary Information. If data can only be shared on request, please explain why in your Data Availability Statement, and also in the correspondence with your editor. Please note that for some data types, deposition in a public repository is mandatory - more information on our data deposition policies and available repositories can be found below:

<https://www.nature.com/nature-research/editorial-policies/reporting-standards#availability-of-data>

Link Redacted

Sincerely,

Melina

Melina Casadio, PhD
Locum Chief Editor, Nature Structural & Molecular Biology
ORCID ID: <https://orcid.org/0000-0003-2389-2243>

Reviewers' Comments:

Reviewer #1 (Remarks to the Author):

The authors have satisfactorily addressed all of my concerns. I am now in strong support of publication of the manuscript.

Reviewer #2 (Remarks to the Author):

The authors have done a great job responding to my concerns and I recommend the paper for publication in its current state.

Version 4:

Decision Letter:

Our ref: NSMB-A49966D

3rd Jul 2025

Dear Dr. Mali,

Thank you for submitting your revised manuscript "Molecular basis for the activation of outer dynein arms in cilia" (NSMB-A49966D) once more. It has now been seen by Rev#1 and their comments are below. The reviewer finds that the new data are strong and support the conclusions, and therefore we'll be happy in principle to publish the manuscript in Nature Structural & Molecular Biology, pending minor revisions to comply with our editorial and formatting guidelines.

We are now performing detailed checks on your paper and will send you a checklist detailing our editorial and formatting requirements in about 1-2 weeks. Please do not upload the final materials and make any revisions until you receive this additional information from us.

To facilitate our work at this stage, it is important that we have a copy of the main text as a word file. If you could please send

along a word version of this file as soon as possible, we would greatly appreciate it; please make sure to copy the NSMB account (cc'ed above).

Thank you again for your interest in Nature Structural & Molecular Biology. Please do not hesitate to contact me if you have any questions.

Sincerely,

Melina Casadio, PhD
Locum Chief Editor, Nature Structural & Molecular Biology
ORCID ID: <https://orcid.org/0000-0003-2389-2243>

Reviewer #1 (Remarks to the Author):

I think the new additions look good, and think the interpretations are sound. I have no concerns.

Version 5:

Decision Letter:

19th Aug 2025

Dear Dr. Mali,

We are now happy to accept your revised paper "Molecular basis for the activation of outer dynein arms in cilia" for publication as an Article in Nature Structural & Molecular Biology.

Your paper will be published online soon after we receive proof corrections and will appear in print in the next available issue. You can find out your date of online publication by contacting the production team shortly after sending your proof corrections.

Authors may need to take specific actions to achieve compliance with funder and institutional open access mandates. If your research is supported by a funder that requires immediate open access (e.g. according to [Plan S principles](https://www.springernature.com/gp/open-science/plan-s-compliance) or the [NIH public access policy](https://www.springernature.com/gp/open-science/us-federal-agency-compliance)) then you should select the gold OA route, and we will direct you to the compliant route where possible. Because authors warrant under our subscription licensing terms that they haven't committed to licensing any version of their article under a licence inconsistent with the terms of our agreement – including the applicable embargo period – publication under the subscription model isn't suitable for authors whose funders require no embargo.

Sincerely,

Melina Casadio, PhD
Locum Chief Editor, Nature Structural & Molecular Biology
ORCID ID: <https://orcid.org/0000-0003-2389-2243>

The authors thank all three reviewers for their helpful comments. Please see our responses to the comments and a detailed description of all changes below.

Reviewer 1 [Referee expertise: dynein]

This is an excellent manuscript with very clearly stated hypotheses and satisfying answers to the questions posed. In brief, the authors follow up on a 2021 manuscript in which they initially described Shulin's biochemical and biological activity along with a high resolution cryo-EM structure. They speculate that DNAAF9 is a likely homolog to the *T. thermophila* Shulin protein, which their previous work demonstrated was an important effector of OAD's autoinhibited state. They find that DNAAF9 localizes to the developing cilia of human airway epithelial cells (but not to mature cilia), co-IPs with OAD components (eg. DNAI2), and its addition to purified OAD causes a change in conformation to a 'closed'/presumably autoinhibited state. The authors then focus on testing the hypothesis that the Arl3 GTPase releases ODA from DNAAF9 in a GTP-dependent manner. They find Arl3 from human and *T. thermophila* both bind to DNAAF9/Shulin. Using negative stain EM and AF2, they map the approximate contact point between Arl3 and DNAAF9 that they confirm via mutagenesis and binding assays. Finally, the authors show that a GTP-bound Arl3 indeed has the capacity to dissociate a *T. thermophila* Shulin-ODA complex. I strongly support publication of the manuscript providing they address a few concerns I have, which may help to improve the manuscript prior to publication.

Major concerns:

1) With regard to "Taken together, these findings indicate that human DNAAF9 interacts with mammalian ODAs and maintains them in a closed conformation sharing its role as an inhibitor with its *Tetrahymena* ortholog Shulin", I wonder why the authors didn't test DNAAF9 in a gliding assay as they did previously (Mali et al., Science 2021)? Although the negative stain EM evidence is fairly strong, I'm not sure their findings "indicate" a role for DNAAF9 in inhibiting ODAs. Such an experiment would provide much stronger evidence that DNAAF9 indeed performs an analogous role to Shulin in ODA function. It would also provide support for the "inhibitory role" proposed in this sentence from the Discussion: "...we favour a model where the critical inhibitory role played by Shulin (DNAAF9) is central to stabilising IFT-ODA interactions for efficient transport into cilia." Without such evidence, the language would need to be changed to reflect this.

This is an important point and although we agree that functional assays such as a gliding assay would directly support DNAAF9's inhibitory role on mammalian ODAs, currently it is

technically impossible to conduct such assays using mammalian ODA motors. Two key limitations prevent this.

- a) Obtaining functionally intact mammalian axonemal ODA motor complexes of sufficient purity and in sufficient quantities remains a major technical bottleneck in the dynein field. Hence, all displacement experiments were performed using purified *Tetrahymena* ODAs.
- b) Methods for reconstituting axonemal dynein's using recombinant subunits obtained from heterologous expression systems (ex. insect cell-baculovirus) have not been developed. We think this is due to the still poorly understood chaperoning requirements that appear to be specific for the biosynthesis of axonemal dyneins.

To reflect this limitation, we have changed the language by focusing the last paragraph more on Shulin and added the below sentence (**lines 432-435**).

“Although our observation that pig ODAs adopt a closed conformation in the presence of DNAAF9 could hint at its inhibitory role in mammalian ODAs (similar to Shulin in Tetrahymena ODAs), more functional assays using mammalian ODAs are needed to directly test this in the future.”

2) Although I'm okay with the model in Figure 2i being presented as is, the data to support it is a bit thin (quantitative MS data). In particular, the model for dynein-1 transport of DNAAF9, although intriguing, may need more support. Do the authors see any cargo adaptor proteins in the MS data? Although not required, this would make this argument more convincing. Also, have the authors tried looking for dynein-1 complex components in their EC vs MC pull-downs?

We do not detect known dynein-1 cargo adaptors in the dataset. However, we detect dynein-1 light intermediate chain (DYNC1LI2) and intermediate chain (DYNC1I2) subunits in EC vs MC IPs with differential enrichments. These are shown in the scatter plots (**Fig. 2a, b**) and the dot plot (**Fig. 2f**). We have amended the text as below to emphasize that the connection to dynein-1 transport needs further support (**lines 223-225**). We have also added question marks in Fig. 2e and 2i to emphasize this point and updated the figure legend accordingly.

“ODAs then reach the peri-basal body pool. Apical enrichment around the basal body could be achieved via the dynein-1 transport machinery or another mechanism. More work is needed to address this”

3) I was a little confused by the back-and-forth between model systems. I think it's great that the authors included all these data, but my head started to spin by the last figure in particular. It would help if there was more explanation as to why they performed some experiments with

human components, and others with *T. thermophila* proteins. For example, why didn't the authors test whether human Arl3 induces dissociation of the DNAAF9-ODA complex?

We performed cross-species interaction studies to highlight that the interaction between DNAAF9 and ARL3 is evolutionarily conserved. Technical limitations of obtaining human/mammalian ODAs restricted us to using *Tetrahymena* ODAs for the reconstitutions and subsequent displacement experiments. These were performed using *Tetrahymena* proteins. We have included the below sentence to explain this point (**lines 875-876**).

“Displacement assays were performed using ODAs purified from isolated Tetrahymena cilia and Tetrahymena Shulin due to technical limitations in purifying mammalian ODAs”

To address the point of going back-and-forth between model systems (and also point #2 raised by reviewer #2) more thoroughly, we performed new reconstitutions under 1mM GTP or 1mM GDP conditions as detailed below:

1. Human DNAAF9 + Human ARL3^{Q71L}
2. Human DNAAF9 + Human ARL3^{T31N}
3. Human DNAAF9 + Human ARL3^{Q71L-FYY}
4. *Tetrahymena* Shulin + *Tetrahymena* Arl3^{Q70L}
5. *Tetrahymena* Shulin + *Tetrahymena* Arl3^{T30N}
6. *Tetrahymena* Shulin + *Tetrahymena* Arl3^{Q70L-FYY}

These new data are consistent with our previous finding of a cross-species interaction between human DNAAF9 and *Tetrahymena* Arl3. We have incorporated these new data into updated **Fig. S10** and updated the text describing these data (**lines 270-272**).

Minor points:

1) Although I was able to guess which was which, the AF2 models for Shulin and DNAAF9 in Fig. S1 are not labeled. Please add labels.

We have now labelled these in **Fig. S1**.

2) It seems that the sequence for *T. thermophila* Shulin is missing from Figure S2. If not, please label accordingly. If so, I think this would be a helpful point of comparison.

The sequence for Shulin was labelled with its UniProt ID (Q22YU3_TETTS). We have re-labelled to include the name Shulin and added the names with corresponding Uniprot IDs in the figure legend.

3) The data presented in the Discussion should either be moved to the Results section (and described in detail) or removed from the paper entirely (and saved for a future paper). This includes Fig. S9 and 10.

We have retained the data showing the hydrocephalus phenotype as it supports a role for DNAAF9 in motile cilia functions. Text relating to this data has been incorporated into the main results section (**lines 192-200**) and updated in the discussion (**lines 402-403**). **Fig. S6** replaces the old Fig. S9. **Fig. S7** replaces the old Fig. S10 (detailed in point 4 below). The numbering of subsequent supplementary figures has been updated accordingly.

4) The paragraphs in the Discussion describing Oda16 and potential transport adaptors (line 366-390) seemed off-topic and was a distraction to me as a reader. I had trouble finding out how this fit into the paper. I feel the same is true for their argument that DNAAF9 is a transport adaptor for IFT. I would suggest removing this part of the discussion and accompanying data in Figure S10 (save it for another paper) and simply focus on discussing the data within the results section.

We have removed the Oda16 data and text describing potential transport adaptors from the manuscript (previous lines 366-390). We have retained the IFT binding data as it validates the interaction with IFT74 and IFT81 detected by endogenous IP-MS. Text relating to this data has been incorporated in the main results section (**lines 213-219**) and the discussion (**lines 409-413**). **Fig. S7** replaces the old Fig. S10.

5) "Given these emerging links between ARL3 and the biogenesis of motile cilia, we recommend that primary ciliopathy patients carrying ARL3 variants should be additionally assessed for clinical features indicative of defective motile cilia." Maybe this is born from ignorance, but wouldn't primary ciliopathy patients already be assessed for potential clinical features of defective motile cilia?

We have rephrased this sentence as below (**lines 423-424**).

"Further work is needed to dissect the emerging links between ARL3 and its effector proteins in the trafficking of motility-related cargoes during motile cilia formation."

Reviewer 2 [Referee expertise: dynein, structural biology]

Issa et al. set out to determine the functional consequence of the interaction between DNAAF9 and the small g protein Arl3. The authors find that DNAAF9 (and its tetrahymena homolog Shulin) preferentially bind to GTP-bound Arl3. Arl3-DNAAF9 interaction displaces DNAAF9 from outer dynein arm complex, thus resulting in dynein activation in cilia. The motivation for this work is important, as ciliary beating defects causes ciliopathies and the findings were described clearly. Pending two important points below, I recommend this work for publication.

Major points:

1. The authors use mutagenesis to try to validate the predicted interface between DNAAF9 and Arl3 (they make three mutations: F51A, Y77A, and Y81A). However, these mutations should be made in the Arl3-Q70L GTP-locked background, not the WT background. The data in Figure 3A shows that WT Arl3 that can GTP cycle doesn't really bind DNAAF9 (maybe there is a little shoulder that shows complex co-elution, but the interaction is not robust enough to see Arl3 on the SDS-PAGE gel in b). The authors should test if the Arl3- F51A, Y77A, Y81A mutant in the Q70L background has reduced affinity for DNAAF9.

This is an important point. To address this, we reconstituted DNAAF9 with human ARL3^{Q71L} or ARL3^{Q71L-FYY} and Shulin with *Tetrahymena* Arl3^{Q70L} or Arl3^{Q70L-FYY}; (FYY = interface triple mutant on a QL GTP-locked background). These data showed that the FYY interface triple mutants on GTP-locked QL backgrounds have reduced affinities for DNAAF9 and Shulin proteins respectively compared to the QL variants alone in both species. We have included these new data in **Fig. 4 (new panel e)** and **Fig. S10**.

2. The data in Figure 5 is very important to the authors model but is the least convincing in the paper. The following points are important to fortify this data:

a. The data in Figure 5d must be quantified, rather than just shown qualitatively. For example, the authors could measure the intensity of the bands of ODA heavy chain: Shulin in peak complex fractions with and without inclusion of Arl3. In fact, it is actually very challenging to tell if inclusion of Arl3 really does reduce Shulin-ODA interaction because there is significantly

more total protein loaded on the gel for the experiment without Arl3.
b. The GTPase dead Arl3 mutant or the Arl3 mutant variants that don't bind (but in the GTP-locked background) would be a great negative control for this study.

We conducted new reconstitution, and displacement experiments and quantified these data using gel densitometry. These have now been included in a new **Fig. 5**. The original Fig. 5 has been replaced by a new **Fig. 6** which presents our overall proposed model. We have also updated the relevant methods section (**lines 875-888**). Briefly, we reconstituted *Tetrahymena* ODAs with Shulin. The reconstituted ODA-Shulin complex was divided equally into two fractions to test the impact of Arl3 in displacing Shulin. The ODA-Shulin complex fractions were incubated with either Arl3^{Q70L} (active GTP-locked variant) or Arl3^{Q70L-FYY} (interface mutant on a QL background) as a negative control. Gel densitometry analyses were performed on replicate runs to calculate the Shulin:ODA heavy chain and Shulin:ODA IC2 ratios for each of the three experimental conditions.

These new data and analyses (included in new **Fig. 5**) show that reduced levels of Shulin co-elute with the ODA complex in the presence of the Arl3^{Q70L} variant compared to in the presence of Arl3^{Q70L-FYY} mutant which has reduced affinity for Shulin. Overall, this suggests that Shulin gets displaced from ODAs by Arl3^{Q70L} (i.e., active Arl3) and is sequestered in a complex with Arl3^{Q70L}.

c. Finally, the authors should show the SEC traces for the experiments with and without Arl3, as well as show gels that have the exact same fractions (and where they correspond to the SEC elution profile) for each experiment. It is hard to assess the quality of this data without seeing the traces and all the same corresponding fractions.

Gel filtration traces for reconstitution and displacement experiments with corresponding gels are included in new **Fig. 5**.

Minor points:

1. Zoom in on figure 1d would be helpful to show the DNAAF9 staining on the cilia.

We have added zoomed panels to show the DNAAF9 and DNAI2 staining in cilia in **Fig.1d** and updated the figure legend.

2. Can the authors report the % open and % closed ODA conformations with and without DNAAF9 (data in figure S5) to quantify the effect of DNAAF9 on ODA structure?

We have added the percentages for open and closed conformations for ODA alone and for ODA mixed with DNAAF9. Text in the legend for **Fig. S5** and the relevant methods section (lines 901-907) has been updated.

Reviewer 3 [Referee expertise: cilia]

The manuscript by Issa et al. from the Mali lab addresses a fundamental question in ciliary biology. More specifically, how are the multimeric axonemal dyneins assembled and maintained in the cytoplasm in an inactive form and then activated once they enter cilia and flagella and attach to the outer doublet microtubules. In previous work the authors identified a dynein chaperone known as Shulin that maintains outer dynein arms (ODAs) in an inactive conformation in the cytoplasm and co-localizes with ODAs in regenerating *Tetrahymena* cilia. In the present study they ask (1) whether this inhibitory mechanism is conserved in Shulin orthologs such as the human DNAAF9, (2), whether it interacts with transport factors to target ODAs to cilia, and (3) how is Shulin/DNAAF9 released from ODAs inside cilia to allow ODA activation?

Most significant findings:

Other studies had shown that DNAAF9 could be pulled down with the small GTPase ARL3, which suggested that DNAAF9 might be involved in the ciliary transport of ODAs. The authors used AlphaFold 2 (AF2) analyses to compare the structures and sequences of Shulin and DNAAF9 (Figures S1, S2). These analyses identified several conserved domains with significant structural similarity that could be involved in binding to ODAs during transport. The authors test this model using human airway epithelial cells cultured at an air-liquid interface to induce differentiation of basal cells into multiciliated cells (Figure 1). Consistent with their hypothesis that Shulin and DNAAF9 share a similar function, they observed that DNAAF9 enters growing motile cilia in differentiating airway cells (Fig 1d, e). Similar results were observed using zebrafish embryos (Figure S3).

To gain a better understanding into the mechanism by which DNAAF9/Shulin might facilitate the transport of ODAs, they used immunoprecipitation, TMT labeling, and mass spectrometry to identify interacting proteins (Figure 2, Supplemental Datafile 1). This identified more than 87 proteins found at different stages of differentiation, including ODA subunits, IFT-B subunits, ciliary tip proteins, and ARL3. The proteomics work appears to be rigorous, and the major take-homes are nicely summarized in their Figure 2, but I would have liked to have seen more

detailed explanation of their results in the Supplemental Datafile 1. If I understand their methods correctly, the IPs were performed using total cell lysates and not purified cilia. Thus, the model proposed in Figure 2i is an inference based on the results from total cell lysates. However, they test several key points of their model using a clever series of in vitro binding experiments.

To test the hypothesis that ARL3 might regulate that activity of DNAAF9 and Shulin, they expressed recombinant DNAAF9 and Tetrahymena ARL3 in its GTP and GDP mimicking states using Q70L and T30N variants respectively. They found that all Arl3 variants co-eluted with DNAAF9 in high GTP (Fig. S6) but only Arl3Q70L binds DNAAF9 in the presence of GDP (Figure 3). Using this Arl3 variant as bait and lysates of Tetrahymena cells undergoing ciliary regeneration, they co-immunoprecipitated several known effectors of Arl3 and Shulin (Figure 3), suggesting that Arl3 is likely to regulate transport of ODAs through its interaction with Shulin.

The authors extended this work by analyzing the interaction between recombinant DNAAF9 (or Shulin) in complex with ArlQ70L in vitro by negative stain electron microscopy (Figure 4, Fig. S7). Single particle averaging identified a subset of images (~50%) containing a DNAAF9/Shulin-Arl3 complex, with the ARL3 subunit bound to the N1 domain of DNAAF9/Shulin. Analytical chromatography confirmed that this interaction was most stable using the Arl3Q70L variant. They extended these observations using AF2 to model the interaction interface and then disrupted the interaction using specific ARL3 mutations (Figure 4e). Further modeling using AF3 suggests that ARL3 interacts with DNAAF9 in the presence of GTP but not GDP (Fig. S8). Additional in vitro studies demonstrate that ARL3 can disrupt the interaction between purified Shulin and ODAs in the presence of GTP (Figure 5).

Overall Evaluation and Significance

In summary, the authors combine sophisticated modeling, well executed proteomics, and elegant in vitro studies to test a new model of how the ARL3 GTPase might regulate the interaction between a dynein chaperone and the ODAs to maintain the ODAs in an inactive state in the cytoplasm and release and activate the ODAs upon entry into the cilia. The experimental logic is clearly described, the experiments are well executed, and the suggested mechanism of regulation for dynein activation in vivo is a novel, exciting hypothesis. Future in vivo studies will be needed to determine the precise timing and location of ODA activation, but the work presented here is an exciting step forward. I think the work will be of broad interest

to people working in the field of motor activation and targeting, and I highly recommend publication.

We thank the reviewer for their helpful overall evaluation of our manuscript.